# Bioinspired handheld time-share driven robot with expandable DoFs

Yunjiang Wang[1,4], Xinben Hu [2,3,4], Luhang Cui[1], Xuan Xiao [1], Keji Yang [1], Yongjian Zhu [2,3,5] ✉ & Haoran Jin [1,5] ✉

Handheld robots offer accessible solutions with a short learning curve to enhance operator capabilities. However, their controllable degree-of-freedoms are limited due to scarce space for actuators. Inspired by muscle movements stimulated by nerves, we report a handheld time-share driven robot. It comprises several motion modules, all powered by a single motor. Shape memory alloy (SMA) wires, acting as "nerves", connect to motion modules, enabling the selection of the activated module. The robot contains a 202-gram motor base and a 0.8 cm diameter manipulator comprised of sequentially linked bending modules (BM). The manipulator can be tailored in length and integrated with various instruments in situ, facilitating non-invasive access and high-dexterous operation at remote surgical sites. The applicability was demonstrated in clinical scenarios, where a surgeon held the robot to conduct transluminal experiments on a human stomach model and an ex vivo porcine stomach. The time-share driven mechanism offers a pragmatic approach to build a multi-degree-of-freedom robot for broader applications.

Biologically inspirations offer innovative approaches to enhance robot capability[1]. By learning from creatures, robots aim not only to execute precise motion and stable force, which they are inherently adept at, but more importantly, to interact with delicate objects, humans, and unstructured environments[2–5], and achieve superior outcomes within constrained spaces. The diversity found in nature has spurred the development of novel robots, diverging from conventional industrial robot frameworks for wider applications. For example, soft robots[6–8], inspired by octopuses, worms, and starfish, offer both conformability and complex motions. These robots leverage their lightweight design and reduced energy consumption to exhibit unique locomotion abilities, navigating rocky or unstable surfaces. Continuum robots[9–12], inspired by the snake or elephant trunks, can change shape and position their tips by flexing along their entire length, making them well-suited to pass through narrow spaces for examination or operation. However, it is costly for robots to replicate the flexible motion abilities of creatures. To be specific, the spine of a snake consists of hundreds

of vertebrae, each functioning as an individual motion unit. This complex structure affords snakes a wealth of degrees of freedom (DOFs), enabling various flexible movements such as lateral undulation, sidewinding, concertina, rectilinear, and slide-pushing[13]. For robots seeking to enhance DOFs by adding more motors, this typically results in significant increases in material costs, control complexity, and bulky robot sizes. Bioinspired actuators[14–16] offer a potentially low-cost approach to expand DOFs but at the sacrifice of the inherent precision and consistency that motors provide.

Snakes are unique vertebrates exhibiting exceptional flexibility. Their spinal cord traverses the vertebral canal, branching spinal nerves at every level to connect muscle fibers. Concurrently, major distributing arteries carry blood pumped from the heart, branching into successively smaller vessels that culminate as capillaries, supplying nutrients to muscles. Stimulated muscles engage multiple segmental spines, enabling multi-DOF motion. Inspired by snakes, we reported a novel time-share driven robot (Fig. 1A) with expandable DOFs by

[1]Key Laboratory of Fluid Power and Mechatronic Systems, Department of Mechanical Engineering, Zhejiang University, 310058 Hangzhou, China. [2]Department of Neurosurgery, Second Affiliated Hospital of Zhejiang University School of Medicine, 310009 Hangzhou, China. [3]Key Laboratory of Precise Treatment and Clinical Translational Research of Neurological Diseases, 310005 Hangzhou, China. [4]These authors contributed equally: Yunjiang Wang, Xinben Hu. [5]These authors jointly supervised this work: Yongjian Zhu, Haoran Jin. ✉e-mail: neurosurgery@zju.edu.cn; jinhr@zju.edu.cn

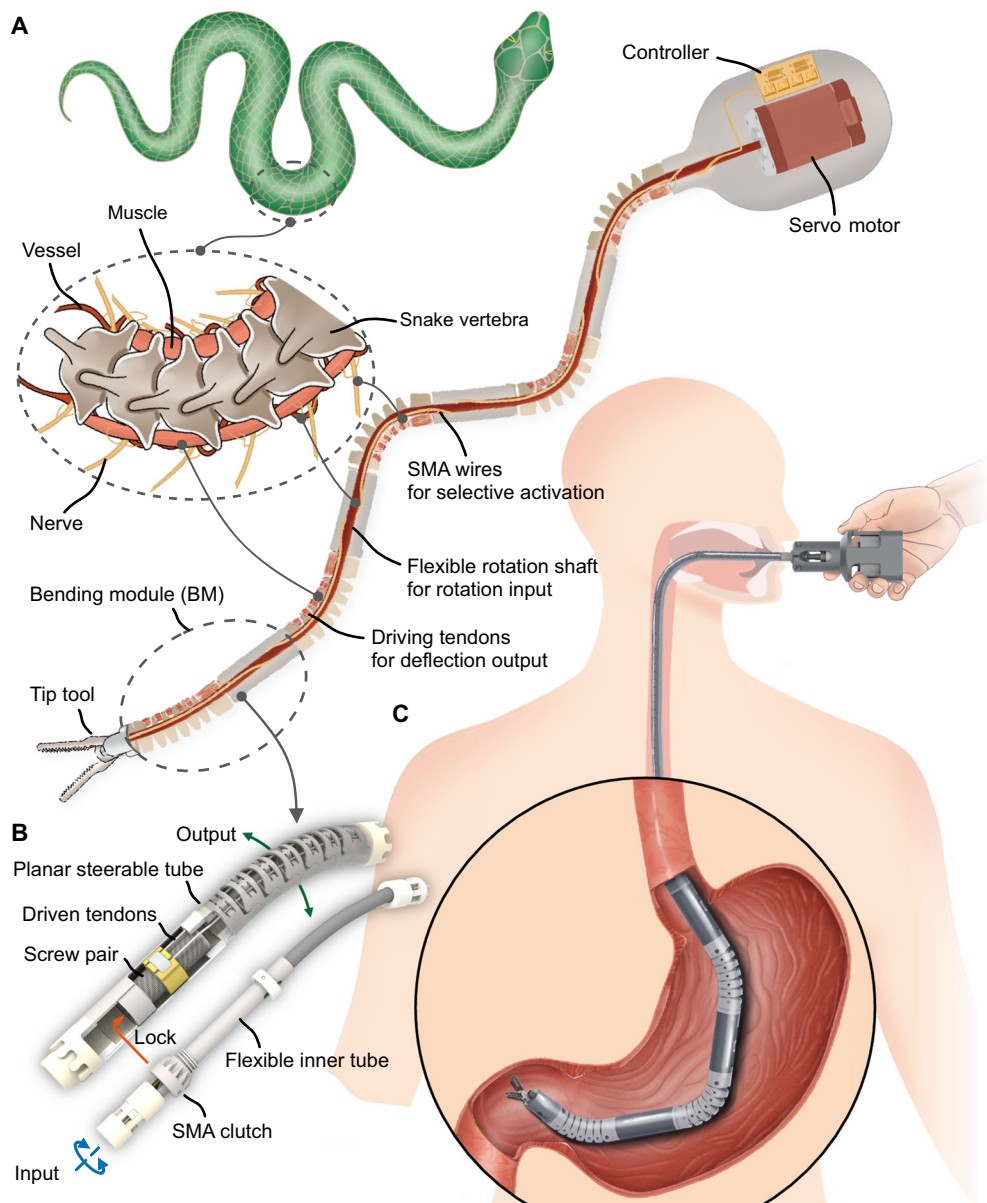

**Fig. 1 | Schematic illustration of the proposed bioinspired time-share driven robot. A** Design of the snake-inspired time-share driven robot. **B** Schematic of one bending module (BM). The screw pair, locked to the shape memory alloy (SMA) clutch, facilitates the conversion of rotational motion into translation, manipulating the elastic tendon to steer this BM. **C** Illustration of the active steer-ability of a handheld time-share-driven continuum robot within a human stomach.

utilizing a single motor to actuate numerous serially connected bending modules (BM). Each BM (Fig. 1B) comprises a planar steerable tube, a flexible inner tube, and an embedded shape memory alloy (SMA) clutch. The planar steerable tube, containing a section of discrete-jointed links, akin to the "vertebra", is steered by tendons threaded through these links, functioning as the "muscle". The SMA clutch, metaphorically described as the "nerve", is activated by a vol-tage signal, securely fastening the relevant components within the planar steerable tube to enable its bending motion. The flexible inner tubes of all BMs are interconnected, akin to "vessels". Those BMs with planner steerable tubes fastened are actuated by the rotation of flex-ible inner tubes. Much like blood vessels perfused by a single heart, all the flexible inner tubes are powered by the sole motor. The robot is cost-effective, compact, and portable, benefiting from sharing a single high-performance motor. Importantly, the BMs remain independent of the motor, allowing rearrangement and reconfiguration for task-oriented assembly.

Based on this design, minimally actuated continuum robots are especially suitable for minimal invasive flexible surgery[17] (MIFS) where the operational approaches are required to be less constrained by anatomical structures. This application necessitates the creation of lightweight and flexible manipulators with minimum footprint. In recent years, continuum robots[18–21] equipped with multiple DOFs, achieved by compacting more actuated cables/tendons, have shown effectiveness. However, these robots[22,23] require an extensive array of motors situated in the base, leading to an expensive, complex-to-control system that occupies a significant amount of space alongside the surgical bed. Additionally, they pose a sharp learning curve for surgeons. Furthermore, in terms of transluminal surgery requiring nearly non-invasive access and high-dexterous operation on distal surgical sites, multi-DOF steerable manipulators extended by straight tubes are unsuitable for establishing an operation workspace through a tortuous path. Meanwhile, extensions with passive, flexible proximal sections support poor distal maneuvers over a long-distance

transmission. Consequently, surgeons still face difficulties in effectively applying their techniques in the utilization of current devices.

Technically speaking, the stiffness of flexible devices is designed as a compromise between possessing sufficient stiffness for distal propagation and operation and being compliant enough to adapt to the structural constraints along a tortuous path[24]. Addressing this requirement involves the flexible robot reserving its shape and marching, a principle known as the follow-the-leader (FTL) motion[25]. It was initially proposed as an algorithm for continuum robots with extensive DOFs. Lately, various FTL mechanisms[26] based on stiffness methods have emerged in medical devices. To intrinsically follow the leader, a strategy involves two tubes alternating as the leader, stiffening to allow the other to march forward[27]. However, it remains unsettled to deliver surgical instruments to the tip and enable operations with sufficient DOFs. Our snake-inspired design fills up the last piece by introducing a flexible rotation shaft. This shaft, composed of interconnected flexible inner tubes metaphorized as a "vessel", extends to the very tip of the robot, actuating endless BMs on the tip and thereby providing the necessary multitude of DOFs.

Last but not least, the proposed prototype was elaborated as a handheld device to provide surgeons with direct enhancement of certain maneuvers, assumed to be susceptive by surgeons in clinical practice. The robot comprises a sequence of socketed BMs mounted on a handheld motor base, demonstrating superior expandability within a portable size compared to current continuum robots (refer to Supplementary Table 1). The applicability was proved in the clinical scenario involving standard gastric ulcer diagnosis and foreign body removal therapy (see Fig. 1C). The time-share driven robot facilitates remote multi-DOF operations via a transluminal approach, presenting a promising and straightforward solution to advance robotic applications in MIFS.

## Results

### Principles and design of the bending unit

To enable a robot to manage numerous DOFs using a single motor, the solution can be distilled into three principles: First, Universal Transmission. The motion input is shared and transmitted along the manipulator. Second, State Keeping. Unactuated modules can withstand external loads. Third, Selective Response. Each motion module independently adopts the motion input. In the proposed BM, the flexible inner tube rotates along the manipulator while adapting to the bending curve, following the first principle. This adaptive rotation is then converted into translation via a hollow screw pair. Subsequently, this translation, appearing as a pull-push motion offset from the BM's neutral axis, translates into bending output through a tendon-driven continuum structure (Fig. 1B). The introduction of a low-lead angle screw pair imparts a self-hold property, preventing external forces from affecting inactive BMs through reverse transmission, following the second principle. Furthermore, the SMA clutch, incorporating a slidable lock, bridges the flexible inner tube and the screw pair. One BM is specified by employing an SMA wire to trigger the slidable lock, following the third principle.

The handheld robot was elongated by a passive extension tube (see Fig. 2A), consisting of a 548 mm stainless-steel braided hose (as the flexible rotation shaft) placed inside of another stainless-steel braided hose (refer to Fig. 2B and Supplementary Fig. 1). Subsequent BMs attached to the passive extension tube derive actuation from the relative rotation between these hoses. Control signals are transmitted to each BM via wires embedded inside the inner hose. The inner hose terminates in a female plug and a male plug, each with 6 pins. A designated pin serves as the common ground wire (GND). The outer hose terminates with a pair of eight-jaw plugs, offering eight phases of connection to adjust subsequent BM's rotation plane.

The capability of the flexible rotation shaft to transmit rotation input along a long distance of curved tube was validated (see

Supplementary Movie 1) on these paired hoses. The optical tracker completed a 360-degree rotation and returned to 0 deg with each turn (refer to Fig. 2C). Zooming into the −5 deg to 5 deg interval on the vertical axis revealed minimal deviations in the 0-degree and 180-degree trials (less than 1 deg). When the bending angle was between 30 deg and 150 deg, the deflected inner hose contacted the inner wall of the outer hose, leading to interference in the inner hose's rotation due to friction. Consequently, the rotation angle fell below 360 deg. Following several rotations, the inner hose's deformation became significant enough to surpass the friction, returning the inner hose to the 0-degree phase. Typically, the deviation remained below 2 deg. The low-lead angle screw pair translated a 2-degree rotation into an approximately 0.0014 mm translation (with a helical pitch of 0.25 mm), indicating a minor influence. The outcome validated the stainless-steel braided hose's capability to accurately transmit rotation. Notably, the friction between the inner hose and outer hose also reduced the inner hose's vibration during rotation. The vibration neared 5 deg when the outer hose remained straight. After the outer hose bent to 180 deg, the vibration was eradicated. Furthermore, the torsional rigidity of the inner hose was tested (refer to Fig. 2D), revealing a slight reduction in torsional rigidity as the outer hose bent. With the outer hose bent to 180 degrees, the inner hose could generate 4 N·mm torque with a 10-degree torsional deformation.

The BM, composed of a planar steerable tube (offering one active DoF), a flexible inner tube, and an embedded shape memory alloy (SMA) clutch, were sequentially attached to the robot base or the end of the passive extension tube. The steerable section of the planar steerable tube was fabricated using laser-profile technology (LPT)[28]. This process involved dismembering a thin-walled stainless-steel tube into a discrete-jointed structure, consisting of interlocked segments. Notches and columns of pits on one lateral side of the tube served as attachment points for the driven tendons (refer to Supplementary Fig. 2). The flexible inner tube contains a section of stainless-steel braided hose (refer to Supplementary Fig. 3), providing torsional resistance to enable flexibility within the planar steerable tube while transmitting torsion along its length. Assembly of a BM entailed sealing one flexible inner tube inside one planar steerable tube (refer to Supplementary Fig. 4). The manipulation of the planar steerable tube within a single BM was accomplished through tendons, a method analogous to the operational principle found in other tendon-driven continuum robots[29–32]. The bending range can be adjusted by altering the fixed point of the driven tendons in the straight section of the planar steerable tube. Consequently, each BM underwent testing to record its specific bending range. The software stored this data to prevent the BM from surpassing its motion range and hitting gears. In Fig. 2E, a single BM underwent testing (see Supplementary Movie 2). The maximum speed of the motor was limited to 2 r/s. Furthermore, the BM successfully sustained a 50 g weight in Fig. 2F, despite each BM weighing only 8.6 g.

In this part, we proposed the BM actuated by a flexible rotation shaft (refer to the inner hose of the passive extension tube). This shaft efficiently transmitted rotation over 548 mm without being affected by the bending shape of the outer hose.

### Performance of the SMA clutch

The material chosen for selectively activating SMA clutches of different BMs is a type of Shape Memory Alloy (SMA) known as nitinol[33,34]. The contraction of nitinol wire occurs with temperature changes[35]. A 150 μm diameter SMA wire, as suggested by Kim et al.[36], was adopted due to its shorter cooling process and splendid durability. To devise a control strategy for the SMA wire and verify its performance, a homemade sliding table (depicted in the left image of Fig. 3A) was developed. The SMA wire contracted, pulling a sliding block against a tension spring (refer to Supplementary Movie 3). The displacement of the sliding block was measured using a micrometer, while a thermal

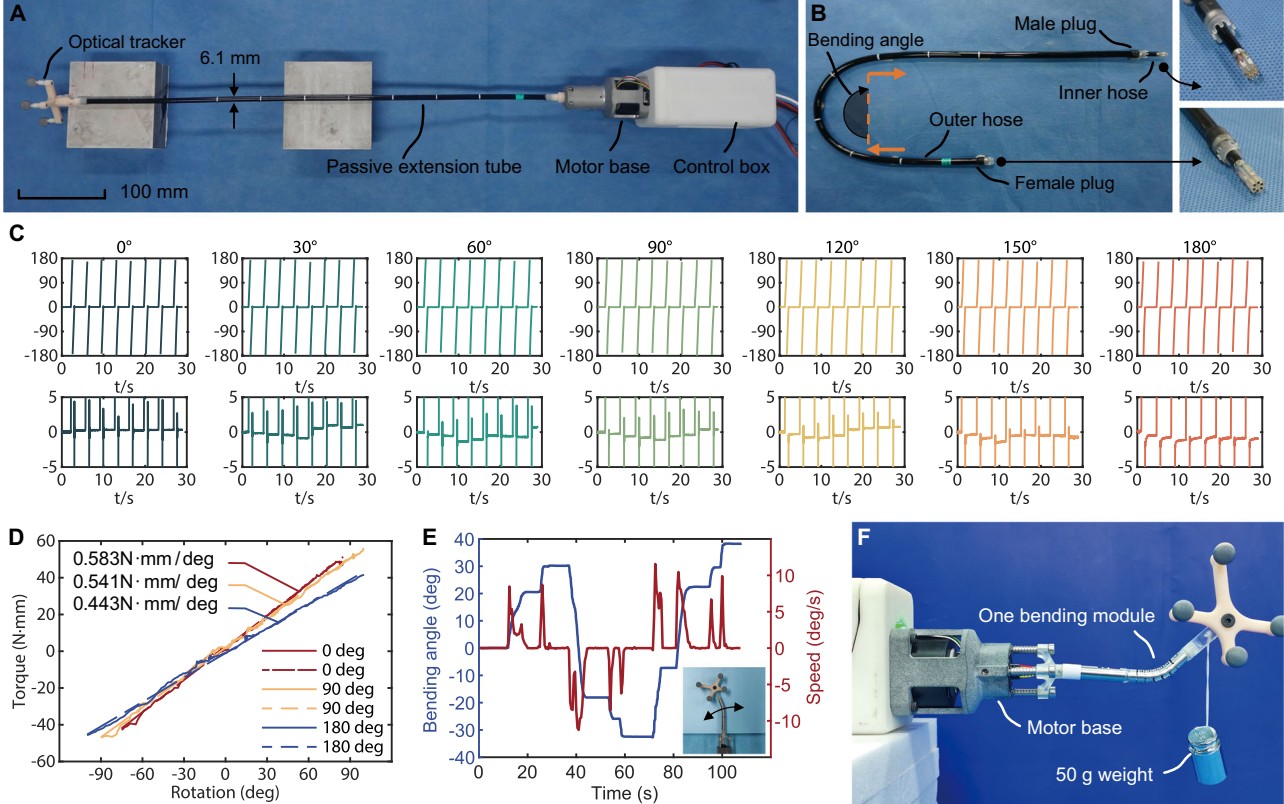

**Fig. 2 | Design and performance of the bending module (BM) actuated by a flexible rotation shaft. A** The prototype of the time-share driven robot with only the passive extension tube attached. **B** The passive extension tube includes two stainless-steel braided hoses with a 3.9 mm hose rotating within a 6.1 mm one. **C** The rotation test of the passive extension tube involved bending the outer hose from 0 deg to 180 deg, with rotation plotted from −180 deg to 180 deg. A zoomed-in view of data between −5 deg and 5 deg was included below each figure. The motor executed an action cycle, rotating once and resting for 4 s. The motor actuated the inner hose from the base, while the free end's rotation was recorded by an optical marker. **D** The torsional rigidity test of the passive extension tube involved rotating the inner hose from the free end side, and torque was recorded from the fixed side. The outer hose underwent straight, 90-degree, and 180-degree bends in three trials. The data were linearly fitted, and the slopes represented the torsional rigidity. **E** One BM performed steering in both directions, achieving a 38-degree bend to the left and a 30-degree bend to the right. The bending angle was plotted in a blue line, while the speed was represented by the red line, reaching 10 deg per second. **F** One BM successfully held a 50 g static weight.

camera recorded the temperature (illustrated in the right image of Fig. 3A). The SMA wire was subjected to varying pulse voltages and durations for heating. The maximum displacement and the corresponding temperature were charted in Fig. 3B. Observations revealed that displacement commenced when the SMA wire reached 40 °C, with no further contraction beyond 50 °C. The results were summarized in three key aspects. Firstly, the maximum displacement achieved was approximately 1.6 mm, equating to a 4% contraction (considering the effective length of the 40 mm SMA wire). Secondly, higher voltage inputs reduced the necessary pulse duration to achieve maximum displacement due to increased heating power. Lastly, it was observed that the pulse should be ceased upon reaching 50 °C, as prolonged heating not only escalated energy consumption but also posed a risk in medical applications due to high temperatures.

Pulse width modulation (PWM)[37] was implemented as a means to curtail energy consumption. The voltage input was substituted with a PWM input featuring five different duty cycles, aimed at sustaining the SMA wire's contracted state. Among the five trials conducted, the one with a 4% duty cycle sustained a relatively low temperature (40 °C) (refer to Fig. 3C). The displacements recorded from these trials are depicted in Fig. 3D. The SMA wire swiftly reached maximum displacement within a few milliseconds. Instances where the temperature remained above 40 °C sustained maximum contraction until the cessation of the PWM input (except for the trial with a 2% duty cycle). The SMA wire subjected to a 4% duty cycle showcased a swifter recovery time back to its original length. These experiments indicate the

possibility of optimizing the contraction performance of the SMA wire by adjusting voltage, pulse duration, and PWM duty cycle.

The functionality of the SMA clutch relies on the SMA wire to toggle the connection between the flexible inner tube and the screw pair, as depicted in Fig. 3E. The SMA wire was folded and securely hooked to a toothed slider. Its two ends were anchored to a fixed ring and linked to two power supply wires. Within the screw pair, a toothed screw facilitated engagement with the toothed slider. The toothed slider was released by a compression spring concealed between itself and the toothed screw, forming a mobile structure referred to as the "SMA clutch". In addition, six more wires were embedded in the flexible inner tube to activate subsequent SMA clutches. Each wire's two ends were connected respectively to the pins of a female plug and a male plug (refer to Supplementary Fig. 3).

To evaluate the SMA clutch, a 4.0 V voltage input was applied for 0.2 s, followed by a PWM input set at a 6% duty cycle—parameters derived from the results of the SMA wire test. As depicted in Fig. 3F and observed in Supplementary Movie 4, the toothed slider efficiently engaged the toothed screw within a mere 0.1 s. Throughout the power-on period (lasting 10 s), the toothed slider maintained a secure connection with the toothed screw. This sustained connection enabled the rotation of the toothed screw controlled by the flexible inner tube, consequently actuating the planar steerable tube. The flexible rotation shaft (see Fig. 2A), extended by the successive flexible inner tubes, reaching all the way to the tip of the robot, selectively engages the corresponding BM for active movement. At last, releasing the

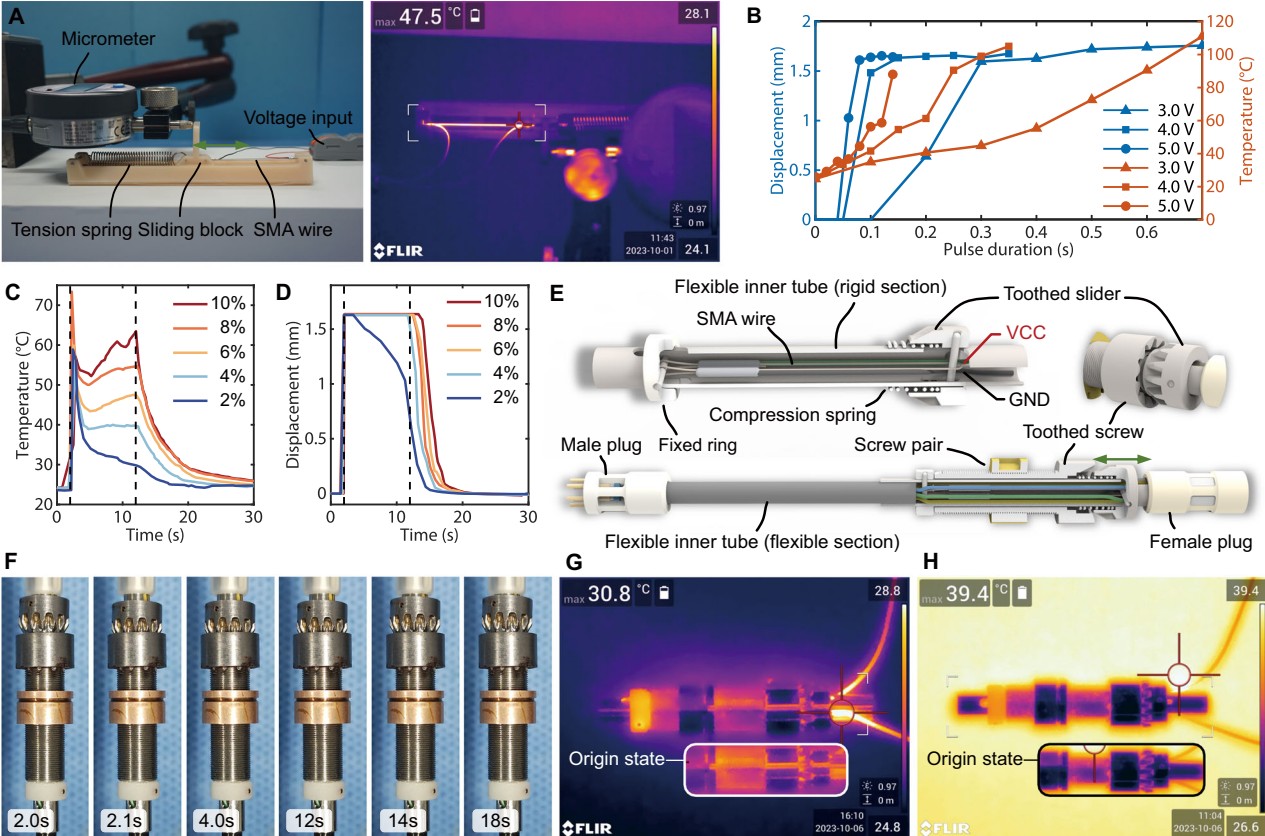

**Fig. 3 | Shape memory alloy functioning as a clutch for selective driving each bending module (BM). A** Left: setup for the SMA wire test; Right: thermal map during the operation of the SMA wire. **B** Maximum displacement and temperature of the SMA wire with varying durations of pulse input. The data comprises three groups with voltage ranging from 3.0 V to 5.0 V. **C** Temperature changes when heated by PWM inputs with different duties (2%, 4%, 6%, 8%, and 10%). Each trial activated the SMA wire with a 4.0 V input for 0.2 s at 2.0 s, followed by powering it with PWM inputs of varying duties. Power was cut off at 12 s, and the cooling process was recorded. **D** Displacement of the SMA wires in tests with different duties. **E** Illustration of the SMA clutch mechanism: Upon heating, the SMA wire contracts, pulling the toothed slider to the left, and locking the tooth slider with the toothed screw. Subsequently, the toothed screw rotates with the flexible inner tube. When the SMA wire cools down, an embedded compression spring resets the toothed slider to its original state. **F** Sequential images showing the working process of the SMA clutch. The SMA clutch was activated at 2.0 s, power cut off at 12 s, and the toothed slider released from the toothed screw at 14 s. **G** Thermal map during the operation of the SMA clutch at room temperature (28 °C). **H** Thermal map during the operation of the SMA clutch at 39 °C.

connection of the SMA clutch took an additional 2 s (refer to Supplementary Movie 4).

Given that the operational temperature of the SMA wire exceeded 40 °C, safety became a critical consideration. We conducted tests to measure the maximum temperature of the SMA assembly (Fig. 3G). Isolated by a bundle of wires and with heat conducted and distributed across two layers of metal tube, the highest temperature was detected on the wire directly connected to the SMA wire. The recorded maximum temperature on the wires was 30.8 °C, which falls within the safe range (refer to Supplementary Movie 5). The wires were further sealed within a plug. Once placed into the planar steerable tube, the SMA clutch is deemed safe in terms of high temperatures. To validate its performance in environments with high temperatures, such as within human organs, the SMA clutch was tested on a constant temperature heating table, as illustrated in Fig. 3H. Even when the environment was heated to nearly 40 °C, the SMA clutch continued to function reliably.

### Locomotion and operation based on a time-share-driven method

The bioinspired time-share driven robot assembly is illustrated in Fig. 4A (also refer to Supplementary Fig. 5). The motor base was positioned beyond a narrow entrance. Extending through the passive extension tube, the distal BMs maneuvered through a tortuous path,

reaching a remote workspace. The distal section comprises four BMs actuated by the flexible rotation shaft within the passive extension tube. The BMs can be assembled in situ, as demonstrated in Fig. 4B (see Supplementary Movie 6). The kinematic model of a single BM and the time-share driven robot is elucidated in the Supplementary materials (refer to Supplementary Figs. 6 and 7).

To demonstrate the capability of the proposed robot to navigate a remote site constrained by a narrow entrance and tortuous path, two experiments were conducted. The first involved navigating through a tortuous path with two bends (Fig. 4C). The robot employed a follow-the-leader motion, with BM 2 leading the way. BM 2 turned left, proceeded to pass the initial bend, then turned right to prepare for the second bend, ultimately completing the route. BM 1 mimicked the leader's motions and successfully traversed the same path (see Supplementary Movie 7). This process was visually depicted in the timeline above Fig. 4C.

The second experiment involved a distal operation to remove a target (Fig. 4D). The robot, equipped with three BMs and a clamping channel at the tip, featured horizontally oriented BMs. The passive extension tube navigated through a narrow opening and a tortuous path to reach the target located at a remote site. Utilizing the time-share driven method, two BMs were activated to successfully extract the target (refer to Supplementary Movie 8).

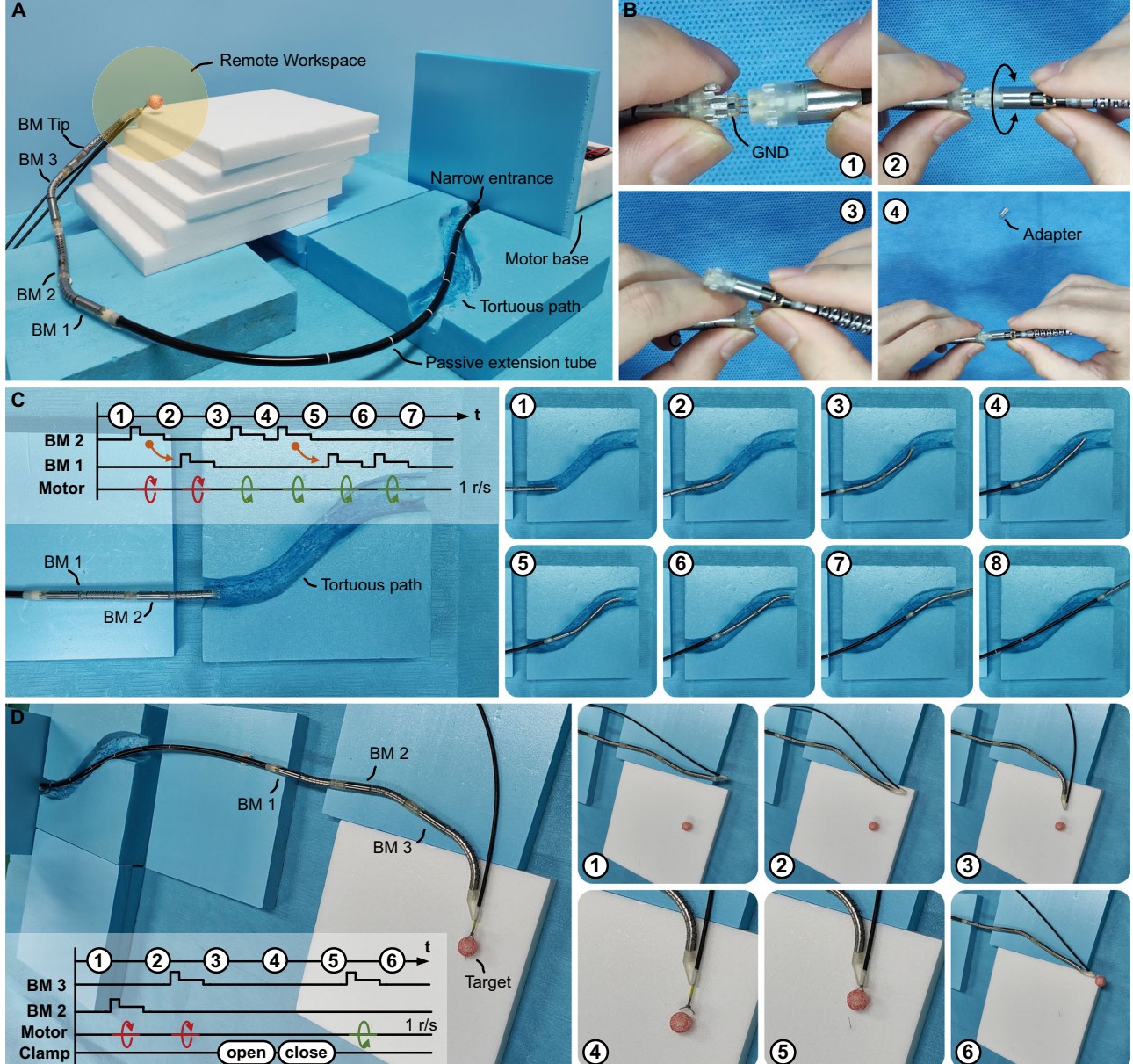

**Fig. 4 | The robot performs locomotion through a tortuous path and distal operation based on the "time-share driven" method. A** The illustration of the time-share-driven robot operating in a remote place. **B** The step-by-step assembly of two bending modules (BM) in situ. Step 1: Connect the flexible inner tubes using an adapter, aligning the GND pins. Step 2: Adjust the bending direction of the next BM's planar steerable tube by rotating it. The plug of the planar steerable tube has eight phases to be matched. Step 3: remove the adapter. Step 4: connect the planar steerable tube directly. Their flexible inner tubes are connected simultaneously. **C** The robot passed through a tortuous path with two bends. Eight top-view snapshots marked 1–8 illustrated the follow-the-leader motion of two BMs. From snapshot 1–3, BM 2 turned left first to pass the first bend, followed by BM 1. Then, from snapshot 3 - 5, BM 2 completed two right turns to pass the second bend. From snapshots 5–7, BM 1 followed BM 2 and performed its two right turns. In snapshot 8, both BMs successfully passed the two bends. **D** The robot performed distal operations through a tortuous path. The distal section consists of 3 BMs marked 1, 2, and 3. BM 3 featured an extended steerable section for a wider bending range. A channel is attached to the tip of BM 3 to send a clamp. Operator-directed activation of BM 2 and BM 3 to reach the red target is portrayed in six snapshots marked 1–6. Snapshots 1–3 show the sequential activation of BMs 2 and 3. Snapshots 4–5 depict the target being gripped by the clamp. Finally, in snapshots 5–6, BM 3 turns and successfully removes the target.

## Diagnosis and therapy experiment of the time-share-driven robot

In a clinical application scenario, we showcased the robot's capabilities through two essential tasks: transluminal examination (diagnosis) in a human stomach model and transluminal operation (therapy) in an ex vivo porcine stomach, both performed by a clinician with extensive experiments in gastroscopy in a simulated operating room setting.

In the diagnostic experiment (Fig. 5A), the time-share driven robot underwent reconfiguration into a three-BM assembly, with a 20 cm

passive extension tube and an integrated camera element (Fig. 5B) to generate an endoscopic view. The robot was delivered from the esophagus into the gastral cavity. The operator kept rotating and inserting the passive extension tube, while the robot deflected for a comprehensive and clear observation of the gastric lining. An external camera was set to record the robot's motion in the cavity from the open window of the model. The examination covered the entire gastric wall, following the route of the greater curvature, fundus, antrum, and upper duodenum (Supplementary Fig. 8). When a lesion was detected,

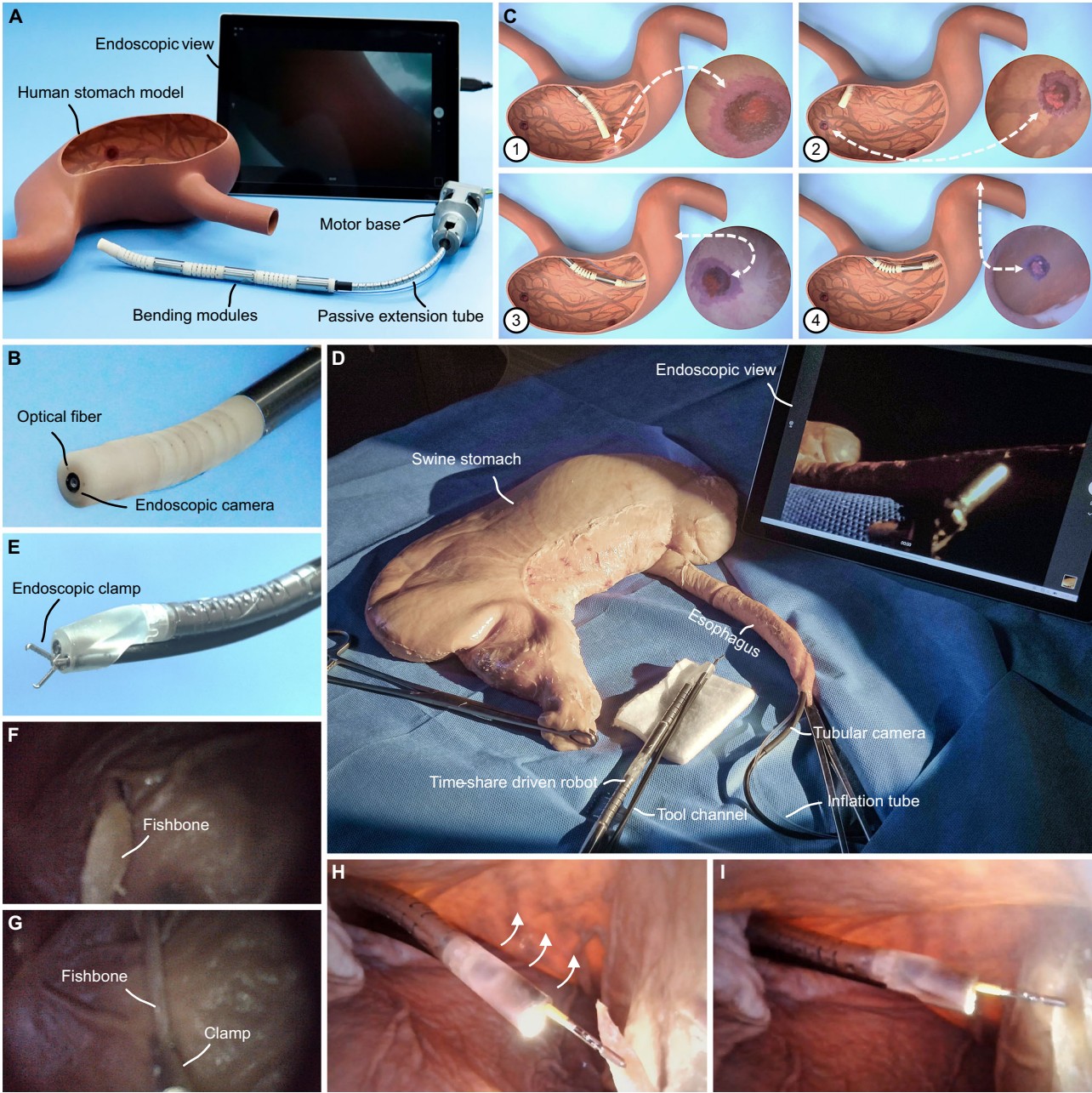

**Fig. 5 | A transluminal diagnosis experiment in a human stomach structure model and a transluminal therapy experiment in an excised swine stomach.** **A** In the scene of the diagnosis experiment, the robot adopted a three-bending-module (BM) assembly with a passive extension tube and a camera element. **B** A camera and lighting element (optical fiber) were integrated at the tip of the robot in the diagnosis experiment. **C** In the diagnosis experiment, the robot moved continuously to center lesions at different anatomical locations. From No. 1 to No. 4 are ulcers on the greater curvature, ulcers on the fundus, ulcers on the antrum, and ulcers on the upper duodenum. The white dashed line showed the lesion location on the model and in the endoscopic view. **D** The scene of therapy experiment in a surgical environment. **E** A tool channel was additionally integrated for endoscopic clamp access. **F** Lesion detection and diagnosis of endoscopic view in therapy experiment. **G** Fishbone removal of endoscopic view in the therapy experiment. **H** Lesion detecting and holding in tubular camera view in therapy experiment, corresponding to **G**. **I** Fishbone removal by manipulator bending in tubular camera view in therapy experiment.

the robot centered the lesion in the endoscopic view for ulcer diagnosis (Fig. 5C). Throughout the procedure, the robot was moving continuously and captured a total of four ulcers at different anatomical locations (see Supplementary Movie 9).

In the therapy experiment (Fig. 5D), the robot was equipped with an additional tool channel (2.4 mm, as shown in Fig. 5E) for accessing an endoscopic clamp (2.0 mm). The inner and outer layers of the passive extension tube were composed of polytetrafluoroethylene, which is acid-resistant, self-lubricating, and harmless to biological

tissues. Polyurethane membranes were used to encapsulate the BMs, which are biocompatible and flexible, making the manipulator water-proof. An ex vivo porcine stomach was properly secured on the operating table and slightly inflated (400 ml) by an air tube to enlarge the inner space (Supplementary Fig. 9). There was a small opening in the fundus for tubular camera access to record the robot's movement in the gastral cavity. A fishbone (5 cm, in Fig. 5F), as a foreign body, was stuck in the lining of the stomach. The robot entered the gastral cavity via the esophagus and began examination in the same route of

diagnosis experiment. After locating the fishbone, the robot adjusted to the planned removal direction at an angle of 30–45 deg from the axis of the fishbone, facing the sharp edge (Fig. 5G, H). The clamp was sent out through the tool channel to hold the bone, and the manipulator bent to slowly lift it in the direction of bone insertion until it separated from the stomach wall (Fig. 5I), ensuring the fishbone tip was away from any tissue (Fig. 5H). Lastly, the bone was taken out by the robot withdrawn (see Supplementary Movie 10).

## Discussion

In this work, inspired by the unique structure of the snake spine and the intricate nerve-muscle-vessel relationship in biology, we proposed a time-share-driven mechanism. This mechanism leverages a single motor to activate multiple BMs. Our approach is based on three fundamental principles—Universal Transmission, State Keeping, and Selective Response. We illustrated this concept through the design of a planar steerable tube, serving as the 'muscle,' a flexible inner tube representing the 'vessel,' and the utilization of a SMA clutch as the 'nerve' for activation. Collectively, we made a prototype consisting of only one motor and multiple BMs that adhere to these principles. These BMs can be fabricated with different motion ranges, assembled in multiple bending plane combinations, lengthened by passive extension tubes, and integrated with various tip instruments, which makes the robot customized and task-oriented. Consequently, this prototype embodies a lightweight, cost-effective, and multi-degree-of-freedom (DOF) robot. It can be handheld, navigates through confined spaces, and establishes a workspace over obstacles.

In recent years, several approaches, referred to as minimally actuated robots, have emerged aiming to reduce the actuation packages while pursuing more DOFs. These ideas focus on matching the actuator with different motion sections along the linear shape of the robot, falling into three distinct categories. The first category, characterized by mobile actuators[38,39], involves repositioning the motor to activate different motion sections. This requires chaining motion sections (forming a serial robot) with a track for actuator locomotion, resulting in lower efficiency when switching between motion sections. The second category, characterized by a central distribution mechanism[40], employs individual transmission mechanisms (usually cable-driven) for each motion section. The cables of transmission mechanisms are centralized to the robot base to be selected by clutches. These robots are cumbersome and show little potential to be compact with a mass of driving cables embedded inside the robot. The third category utilizes tendon-driven continuum robots, employing stiffening/shape-locking methods to alter actuatable sections[41]. Various shape-locking methods have been explored[42], with the SMA clutch being favored for its rapid response time and compact structure[43]. Clutches positioned in local sections enable distributed and modular assembly. However, robots within this category[44] are threaded with driving tendons/cables along their entire bodies, hindering true modular assembly and expandability. Additionally, these sections lack significant independence as the robot is primarily switched between configurations to execute different trajectories. In contrast, our snake-inspired design achieves a minimally actuated robot in a completely modular assembly. The actuator generates standard rotation input, and each motion module independently generates its motion without being restricted by other modules.

Furthermore, we evaluated the prototype's performance in clinical applications, including a diagnosis experiment on a human stomach model and a therapy experiment on a porcine stomach. The robot demonstrated its ability to pass through narrow openings and along tortuous paths toward remote operating sites, showcased in Supplementary Movies 6 and 7, meeting the requirements for transluminal diagnosis. In addition, during the transluminal therapy experiment, the robot exhibited excellent flexibility and maneuverability compared to the gastroscopes. The robot could approach the target with an adjustable angle and lift the bone by bending the manipulator, rather than drag the bone along the endoscope axis as in gastroscopic procedures, minimizing the surgery-related damage to the tissue. These results showed the potential of our robot to be a new solution to handle multi-DOF tasks in difficult-to-reach surgical spots. For clinical applications of the proposed robot, it needs to integrate more channels for suction, inflation, and flushing. Those channels encircle the robot and form a robotic endoscope system with a tube diameter of 1 cm by encapsulation of currently applied gastroscopy, which seals the robot to meet the requirements of internal channel sterilization and external surface sterilization and ensures safe contact with human tissue.

In the future, it is worth embedding sensors in BMs for motion feedback so that closed-loop control can be established for robot-assisted surgery. Meanwhile, motion efficiency can be promoted. The idea of a time-share-driven mechanism sacrificed time efficiency in switching between BMs. It was proved to be feasible that the BMs could move at individual paces to the planned bending angle while the motor kept running to control the speed of movement. Last but not least, although manual control was involved in the diagnosis and therapy experiments, robotic control and tip navigation are feasible under the established kinematic model of a time-share-driven robot (Supplementary Fig. 8). The time-share-driven mechanism provides an unprecedented way to bring the controllable DOFs and the flexibility of the continuum robot to the next level.

## Methods
### Construction
The time-share driven continuum robot comprised two main components: the motor base connected with a control box and several reconfigurable BMs (Supplementary Fig. 7). The base was lightweight (10 cm long, 4.6 cm side length, 202 g weight) housing only one servo motor with a built-in motor drive board and position encoder (MyActuator Technology Co. Ltd, Suzhou, China, RMD-L-4015). In the motor base, the wires for powering up clutches were placed inside the output axle of the motor and connected to a slip ring (MOFLON Technology Co. Ltd, Shenzhen, China, MT0522). The clutch wires and the wires of the motor were connected to a control box with an STM32 controller (DJI Technology Co. Ltd, Shenzhen, China, ROBOMASTER Development Board Type A) inside, which provided more than 16 channels of PWM output and CAN communication protocol to control the motor. An amplifying circuit was utilized for a stable PWM output. The motor base could be directly attached to the control box. The total weight was 494.6 g then.

### Fabrication of the tubes
The discrete-jointed tubes for the planar steerable tube are stainless-steel (SUS304) laser-profiled by a specialized laser cutting machine (Kunshan Yunco Precision Co. Ltd, Jiangsu, China, YC-ETLC8), and the cutting seam width is 30 μm. The patterns are formed by a combination of tube rotation and laser translation. The flexible inner tube and the passive extension tube were stainless-steel braided hoses (Supplementary Fig. 1).

### Fabrication of the SMA clutch
The fabrication of the SMA clutch is illustrated in Supplementary Fig. 3. The SMA wire is the key component. The length is 46 mm. At this length, the resistance of the SMA wire is about 1.8 Ω. The connection between the SMA wire and conduct wires has to be tight and soldered by tin to prevent high resistance. Each SMA wire contracted on the homemade sliding table to adjust the length in a cooled state.

### Time-share-driven control of the manipulator

The SMA clutch responds in 0.1 s (Fig. 3F). To ensure a tough connection, the motor is actuated with a 0.5 s delay after the PWM input starts. The PWM inputs are halted immediately when the motor stops. For the experiments in Fig. 4, only one BM was running at a time. To pursue higher efficiency, the motor could be kept running in speed control mode, and then the motion of each BM was determined by its PWM span.

### Reporting summary

Further information on research design is available in the Nature Portfolio Reporting Summary linked to this article.

## Data availability

The data generated in this study are provided in the Source Data file. Source data are provided in this paper.

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

## Acknowledgements

This work was supported in part by the National Science Foundation of China (NSFC) under Grant Nos. 52275549 (H.J.), and in part by the Key Fund of the Zhejiang Provincial Natural Science Foundation of China under Grant Nos. LZ23E050004 (H.J.). This work was also supported in part by the Key R&D Program of Zhejiang Province under Grant Nos. 2017C03018 (Y.Z.), Nos. 2024C03071 (Y.Z.), and Nos. 2020C01101 (K.Y.). This work was also supported in part by the advanced research program of Donghai Laboratory under Grant Nos. DH-2022ZY0011 (H.J.). We thank Z. Zhang from Hangzhou Risheng Medical Technology Co., Ltd for providing advanced machining technology, H. Lu, and J. Zhang from Hangzhou SKONSIN Health Co., Ltd. for providing a customized endoscope. J. Jing from Zhejiang Provincial People's Hospital for consultation for potential uses in surgery. Laboratory Animal Center of Zhejiang University for providing the standard animal experiment site and consultation.

## Author contributions

Y.W. conceived the time-share driven robot, made the SMA clutch, analyzed the robot motion and experiment data. Y.W., X.H. and L.C. jointly fabricated the bending modules and the motor base. X.H. studied the medical background, designed the phantom and ex vivo experiments, and revised the manuscript. X.H., L.C. and X.X. carried out the experiments and collected the data. X.X. produced the figures and videos. Y.W., X.H. and H.J. co-wrote the manuscript. H.J., Y.Z. and K.Y. supervised the research, reviewed, and edited the manuscript.

## Competing interests

Y.W. and L.C. applied for a patent related to this work in China (nos. 202110438246.7). Y.W., L.C., X.X. and X.H. applied for two patents related to this work in China (nos. 202110672729.3 and 202110672748.6). The other authors declare that they have no competing interests.
