## [Peer Review File · Nature Communications]

REVIEWER COMMENTS

Reviewer #1 (Remarks to the Author):

In this paper, the authors introduced a time-share driven robot which can be held in one hand during the non-invasive and high-dexterous operation. The snake-inspired robot was developed using motion modules powered by only one motor, and satisfies the three driven principles, such as universal transmission, state keeping and selective response. The performance of the robot prototype was demonstrated in the clinical application scenarios, including the human stomach model and ex vivo porcine stomach. In general, this paper provides a promising investigation in the controllable degree-of-freedom development, which can be widely used in the difficult-to-reach surgical applications.

There are several concerns about this paper:

1. Snake-shaped robots are commonly constructed by a large number of short rigid links, and have good plasticity in the multi-degree-of-freedom movements. By considering that the proposed robot can generate a degree of deformation during the movement along object surface, how can it maintain energy stability with only one motor? The authors should explain the energy consumption principle, as well as compare it with other existing continuum robots.
2. Kinematic modelling of robots plays an important role in the movement characteristics analysis. The degree of freedom of each joint in the proposed bioinspired time-share driven robot are coupled together, which increases the complexity of kinematic analysis. It would be better to analyze the kinematics and dynamic models of the proposed robots from the perspective of design principles, and detail the relationship between the kinematic basis and the time-share driven mechanism.
3. As the proposed robot was designed for the clinical applications, safety performance should be our first consideration. There are questions about the sentence mentioned in Page 4: "The material for selectively stimulation is a kind of SMA–nitinol. The contraction of nitinol wire occurs as the temperature changes." Please detail the maximum and minimum values of the environment temperature that would happen in the clinical surgery. Is there any protection if it exceeds the temperature that the protein can tolerate?
4. Currently, many surgical intervention robots are made of flexible polymer materials, and adopt the pneumatically and hydraulically driven mechanisms. The proposed bioinspired robot avoids the commonly used driven mechanisms, which may increase the modeling complexity and response time. Please refer to more relevant literature and highlight the advantages of the proposed robot through experimental comparison.
5. Fig. 2(B) displays the joint shift in the bending progress. Why did the authors set the link lengths as 2.0 mm, 3.0mm and 4.5 mm? Fig. 4(B) shows the contraction rate of the SMA wire using PWM. Please detail the duty value selection in the following sentences:

"Three of the lines were under a constant duty PWM (20%, 50%, 80%). One was from full duty to 20% (100%, 70%, 40%, 20%, 20%...). The left one was from full duty to 50% (100%, 70%, 50%, 50%...)."

6. In the diagnosis and therapy experiments of the proposed robot, the author did not explain the application of three driven principles in the interaction process. For example, how did the "muscle" PSS, "vessel" FIS and "nerves" SMA cooperate to perceive the stomach environment and perform the desired motor behaviors?

7. Please give more information about the corrosion resistance and biosafety of the material in stomach acid, furthermore, also give more analysis of the experimental results of the clinical tests, rather than just stating the results only.

Reviewer #2 (Remarks to the Author):

This manuscript introduces a handheld device that can selectively alter the bending motion of actuation modules using a single motor, through an SMA-based clutch mechanism. The mechanism and structure of the robot are very interesting and addresses the difficulties encountered in developing continuum robots, such as high degrees of freedom in a compact form. The authors also demonstrate its potential for use in medical interventions.

However, the current form of the introduction needs to be rewritten as it presents general comments on robots, bioinspired robots, MIS, and other related topics in a poorly organized manner. In addition, the results section should focus on evaluating the robot's performance extensively, rather than solely introducing fabrication details related to its structure and control.

A rigorous literature review on continuum robots, specifically for minimally invasive surgery (MIS), is recommended. As the key contribution of this paper is believed to be the universal transmission and selective bending control, the authors should compare it with other approaches as outlined below.

1. A Novel Underactuated Continuum Robot With Shape Memory Alloy Clutches, Bishop et al. 2022. doi: 10.1109/TMECH.2022.3179812.

2.. Multiple Curvatures in a Tendon-Driven Continuum Robot Using a Novel Magnetic Locking Mechanism, Pogue et al. 2022. doi: 10.1109/IROS47612.2022.9982193.

3. Continuum robots adopting follow-the-leader mechanism.

Is there any concern regarding the total weight of the bending modules in operation? Could you please provide specifications for a single bending module? The authors have only mentioned the weight of the motor base.

How would the operator know the position of a threaded gear within the stroke of a screw, if it is not visible inside the human body? Is the threaded gear always centered when the system is at rest and without any bending? And how can one avoid hitting the gear at either end of the screw?"

Specific comments:

Title:

In the current version of the title, "expandable dexterity" is used, but the robot described in the manuscript is actually designed for expandable degrees of freedom (DOF) rather than dexterity. The

manuscript does not evaluate the dexterity of the robot. To show dexterity with this robot, the robot should be able to maneuver surgical tools with precision and flexibility.

Line 38-39:

The current form of the robot seems to have less controllability and precision?

Line 54-55:

The surgeon's learning curve may substantially vary the types of user interfaces used for control.

Line 68: State Keeping

Is there any potential issue in "state keeping" with the backlash of the lead screw and mechanical clearance in manufacturing and fabrication process or structural stiffness? Any clearance would be accumulated along the serial connection and affect the end pose of the robot.

Fig 2B: what is joint shift of interlocked tube pieces? It was hard to understand why 4.5 mm value was adopted in the current design through the manuscript.

Line 88-89:

This tube could be bent to a minimum radius of 17.2 mm and was much more rigid to resist out-of-plane forces than the continuum tube.

How was a minimum radius of 17.2 mm derived? Is it from FEA?

It would be helpful to show the stiffness of the structure for each direction or DOF using FEA simulation (e.g. torsional stiffness instead of lifting weight).

Line 92: what is S14?

95: making it suitable for long-distance transmission,

I would concern non-uniform torque transmission at the end. Please evaluate such long distance transmission in terms of torque loss? For example, depending on number of BM modules, external load and bending angles.

102:

Please describe the specification of the super-elastic tendon used in the supplementary.

Was any pre-load applied on fixed tendon?

Line 69-70:

The system assumes no length change in FIS because it lies along the neutral axis of bending. Does the curve of FIS always lie on the neutral axis? If not, it may undergo length change.

Line 113~: The section Third principle - selective response includes,

The amount of detail provided in this section is excessive and would be more appropriate for inclusion in the supplementary materials, such as the instructions on how to wind SMA and control its duty.

Therefore, I suggest that you focus instead on describing how the contraction of the expansible tubular structure is utilized to hold the screw and facilitate rotation with FIS selectively.

Line 118-119:

The 100 μ m SMA wire failed to drive a payload of 1000 g weight.

Why was 1000g payload tested? Is it due to the stiffness of the expansile tubular structure?

Line 122:

As the PWM test depicted in Fig. 4B, the best control strategy is to start with a full-duty PWM to shorten the response time, then drop the duty to a low level (50%) to avoid over-heating.

So, what was the temperature on the SMA element with each control method?

The contraction and recovery (cooling) behavior would be totally different in the human organs.

Line 139-140:

The motor rotated at a speed of 1 r/s. A 6 s PWM output was sent to BM 1, then BM 2, and ended up with an 8 s PWM to motivate both BMs (Movie S3).

Why was 8 s taken for driving the two modules, was it due to loss?

Line 148~

In Diagnosis and therapy experiments of the time-share driven robot section and Movie S5 & S6, one or two BM modules are only shown. It would be better to show how the robot can pass through narrow openings and along tortuous paths (e.g. how the multi-segment robot works with time-share control). Fish bone removal in Movie S6 does not demonstrate the advantage of this robot. It seems to be just operation of an endoscopic clamp.

In Fig 5D, can does the robot appropriately operate on flexible base (extension tube) and keep its shape? Please provide the supplementary video of this remote operation.

Supplementary Materials

1. Comparison of this work with continuum robot designs.

This prototype also entails a large controller back of the motor base. So, it may be unfair the soft robot air/water pumps are heavy.

Reviewer #3 (Remarks to the Author):

General: I find the robot very interesting. However, from the description of the robot in the paper it is very hard for me to understand how exactly it works and what the exact design requirements were. That is a pity because the robot seems quite novel to me. The working principle of the robot requires a much more clear explanation at a functional level. Specifically:

- Introducing many abbreviations in lines 43-51 makes the paper very hard to read and to understand. Use less abbreviations in the paper to make it easier to read.
- Lines 56-57 “the controllable degrees-of-freedom supplied by remotely actuated tendons are consumed quickly in the wandering of obstructive tissues”. No idea what you mean with this “wandering” – further explanation needed.
- Line 58 “surgeons frequently find a target unreachable”. To my knowledge such robots are not yet used by surgeons; they are still in the research phase. So what exactly do you mean here?
- Lines 60-62: can you show photos of the robot used in this experiment, can you give more information about this experiment, with what continuum robot did you exactly compare it? Figure 1 shows only drawings of a robot; it does not show a real robot. So I do not understand how you tested the applicability in a clinical scenario, lines 62-63. Or do you refer in this section to another experiment explained in an already published paper? Add then a reference to the paper where I can find the details of this experiment.
- End of introduction: mention what exactly the goal is of the research presented in the paper. What are the design requirements of the robot in terms of overall diameter, length, force that it can exert on the environment, etc.?
- Lines 83-98 – it is very hard to understand what exactly the problem was here and how you solved it. I do not really understand it. What do you mean with “joint shift” and why exactly was it a problem?
- Lines 99-112: also rather hard to understand. The construction in Fig. 3 seems to be just a segmented steerable joint; it is unclear to me what the “state keeping” feature exactly is, or is it just an electric actuator that stops moving? Requires more explanation. Are the “super elastic” tendons made from SMA? Why not just steel cables?
- Lines 113-130: it is not clear to me how this principle works. What is the purpose of the “expansile tubular structure” in Fig. 4? What was the purpose of the SMA? What is exactly “selective” in this construction? The text zooms in into so much detail about some experiment with the SMA that the overall working principle is not well explained. “the SMA failed to drive a payload of 1000 g”. What was the required payload then? Why is 1 Kg not enough?
- Fig. 4B shows several plots but it is unclear to me what these plots mean and what I can conclude from the figure. Why do you show these plots? What was the design requirement exactly and how can I see from Figs 4A and 4B what solution is best?
- Was the experiment in the stomach carried out by a clinician in a clinical setting?
- The discussion is very technical, add a section about the clinical applicability of the design and issues still to be solved for clinical application, for example dealing with sterilisation, patient safety, biocompatibility, etc.

Response Letter

Reviewer #1

General Comment:

In this paper, the authors introduced a time-share driven robot which can be held in one hand during the non-invasive and high-dexterous operation. The snake-inspired robot was developed using motion modules powered by only one motor, and satisfies the three driven principles, such as universal transmission, state keeping and selective response. The performance of the robot prototype was demonstrated in the clinical application scenarios, including the human stomach model and ex vivo porcine stomach. In general, this paper provides a promising investigation in the controllable degree-of-freedom development, which can be widely used in the difficult-to-reach surgical applications.

There are several concerns about this paper:

Comment 1:

Snake-shaped robots are commonly constructed by a large number of short rigid links, and have good plasticity in the multi-degree-of-freedom movements. By considering that the proposed robot can generate a degree of deformation during the movement along object surface, how can it maintain energy stability with only one motor? The authors should explain the energy consumption principle, as well as compare it with other existing continuum robots.

Response:

Thank you for your advice. Although the proposed robot contains several bending modules. **The bending modules in idle state do not consume energy**, because these bending modules are locked mechanically by the screw pairs. Thus, they support the distal part without energy consumption. In every step, the single motor only actuates those activated bending modules. Also, the activated bending modules take consistent movements, because they are actuated by the same motor.

Compared with other existing continuum robots, the proposed robot is actuated by one flexible shaft. This strategy sacrificed efficiency since the multi-degree-of-freedom movements have to be uncoupled and executed one by one. This sacrifice enables the robot to be compact and modular assembled to face specific scenarios. Also, complex trajectories concerning the compound motion cannot be performed. The proposed robot does not have to balance the energy consumption of multi-degree-of-freedom movements. The current motion pattern is easy to understand by the operator because he only determines the activated bending modules and sends the command. Also, the key merit is that the robot can be actuated from a remote side through a tortuous path. The distal movements are not interfered with by the curvature change of the path. Continuum robots with many DoFs present difficulty in extending their distal part to a remote site and still perform good operation.

We revised the Introduction to clarify the difference between our robots and other existing continuum robots.

-----Manuscript Revised (Introduction / Page 1~2)-----

Based on this design, minimally actuated continuum robots are especially suitable for minimal invasive flexible surgery¹⁶ (MIFS) where the operational approaches are required to be less constrained by anatomical structures. **This application** necessitates the design of lightweight flexible manipulators with minimum footprint. **In recent years,** continuum robots¹⁷⁻²⁰ equipped with multiple DOFs by compacting more actuated cables/tendons have

been proven to be an effective approach. However, these robots^{21,22} are actuated by a large number of motors in the robot base, making the robot system especially huge and expensive, and showing a sharp learning curve to surgeons. Furthermore, in terms of transluminal surgery for nearly non-invasive access and high-dexterous operation on distal surgical sites, the multi-DOF steerable manipulators extended by straight tube are unsuitable to establish an operation workspace through a tortuous path, while extension with passive flexible proximal section supports poor distal maneuvers over a long-distance transmission. Consequently, surgeons still face difficulties in effectively applying their techniques in the utilization of current devices.

Technically speaking, the stiffness of flexible devices is designed as a compromise between being stiff enough to enable distal propagation and operation and being compliant enough to adapt to the structural constraints along a tortuous path²³. To address this, it is required that the flexible robot reserves its shape and marches, stated as the follow-the-leader (FTL) motion²⁴. It was first proposed as an algorithm working on continuum robots with enormous DOFs. Lately, various FTL mechanisms²⁵ have been proposed in medical devices based on stiffening methods. To intrinsically follow the leader, two tubes take their turns to be the leader and get stiffened for the other to march forward²⁶. However, it remains unsettled to send the surgical instruments to the tip and enable its operations with sufficient DOFs. Our snake-inspired design fills up the last piece with a flexible rotation shaft. The flexible rotation shaft, metaphorized as a "vessel", can be extended to the very tip, and actuate endless BMs on the tip, thus providing as many DOFs as demanded.

Last but not least, the proposed prototype was elaborated as a handheld device to provide surgeons with direct enhancement of certain maneuvers, assumed to be susceptible by surgeons in clinical practice. The socketed BMs were installed on a handheld base housing one servo stepper motor. The robot consists of a chain of socketed bending modules (BM) installed on a handheld motor base. The bioinspired handheld time-share driven robot shows expansibility in a portable size over current continuum robots (Table S1 in the Supplementary materials). The applicability was proved in the clinical scenario: standard diagnosis of gastric ulcers and therapy of removing the foreign body (Fig. 1C). The time-share driven robot supports remote multi-DOF operations via a transluminal approach, promising to promote the robotic application in MIFS as a simple and effective solution.

Comment 2:

Kinematic modelling of robots plays an important role in the movement characteristics analysis. The degree of freedom of each joint in the proposed bioinspired time-share driven robot are coupled together, which increases the complexity of kinematic analysis. It would be better to analyze the kinematics and dynamic models of the proposed robots from the perspective of design principles, and detail the relationship between the kinematic basis and the time-share driven mechanism.

Response:

Thank you for your advice. The joints (bending modules) are uncoupled. Each activated bending module is actuated by the rotation of the flexible inner pipe and changes its shape. Its flexible inner pipe gets bent while the rotation transmitted to subsequent bending modules is not influenced by the bending curvature. To prove that the bending curvature would not influence the rotation of the flexible hose placed inside the bent outer hose, we added Movie S1. We also carried several tests presented in Fig. 2 to prove that the rotation can be transmitted along a bent shape with very little lose. Because all the bending modules are actuated by a rotation of a flexible shaft, not effected by the bending curvature, we can focus on the kinematic model of a single bending module and actuate bending modules one by one. In conclusion, the kinematic model is rather simple without coupled movements.

Fig. 2. Design and performance of the BM actuated by a flexible shaft. (A) The time-share driven robot prototype is attached with only the flexible extension tube. (B) The components of the extension tube. The extension tube contains two stainless-steel braided hoses. A 3.9 mm hose rotates inside of a 6.1 mm one. All the subsequent BMs attached to the tube are actuated by the relative rotation between the two hoses. The control signals are transmitted to each BM via the wires embedded inside the inner hose. Two ends of the inner hose are a female plug and a male plug. Each contains 6 pins. One pin is marked as the common ground wire (GND). The outer hose ends with a pair of eight-jaw plugs. They have eight phases of connection to adjust the rotation plane of the subsequent BM. (C) The rotation test of the extension tube. The outer hose was bent from 0 deg to 180 deg. The rotation was plotted from -180 deg to 180 deg. A small figure to zoom in on the data between -6 deg and 6 deg was attached to each figure in below. In each trial, the motor was programmed to repeat an action cycle, in which it rotated one round and rested for 4 s. The inner hose was actuated by the motor from the base, while the rotation of the free end was recorded by the optical marker. (D) The torsional rigidity test of the flexible extension tube. The inner hose was rotated from the free side end and the torque was recorded from the fixed side. The outer hose was straight and bent to 90 deg and 180 deg in three trials. The data were fit with straight lines drawn with dashed lines. The slopes of the lines were marked as torsional rigidity. (E) One BM performed steering to both sides. This BM could bend 38 deg toward the left side and 30 deg toward the right side. The bending angle was plotted in a blue line. The speed was plotted in the red line. The speed reached 10 deg per second. (F) One BM held a 50 g still weight.

Comment 3:

As the proposed robot was designed for the clinical applications, safety performance should be our first consideration. There are questions about the sentence mentioned in Page 4: “The material for selectively stimulation is a kind of SMA–nitinol. The contraction of nitinol wire occurs as the temperature changes.” Please detail the maximum and minimum values of the environment temperature that would happen in the clinical surgery. Is there any protection if it exceeds the temperature that the protein can tolerate.

Response:

Thank you for your suggestion. The temperature is an important issue. We add several experiments to evaluate this aspect. The temperature of the SMA wire can be kept below 50 °C in a working state (Movie S3). After the SMA wire was isolated by several layers of tubes, the heat has little influence on the environment. According to the test in supplementary movie S5, the temperature of the packed SMA wire would cause no harm to the surroundings.

We also explained in the manuscript to eliminate the temperature concerns of the SMA wire.

-----Manuscript Revised (Results / Page 5)-----

Fig. 3. Shape memory alloy works as a clutch for selective driving each BM. (A) Left: the setting for the SMA wire test; Right: the thermal map when the SMA wire was working. **(B)** The maximum displacement and temperature of the SMA wire with different durations of pulse input. The data contain three groups with voltage changing from 3.0 V to 5.0 V. **(C)** The temperature changes when heated by PWM inputs with different duties. The duties of the PWM inputs were 2%, 4%, 6%, 8%, and 10%. In each trial, the SMA wire was activated by a 4.0 V input at 2.0 s which lasted 0.2 s, then powered by the PWM inputs with different duties. The power was cut off in the 12 s and the cooling process was also recorded. **(D)** The displacement of the SMA wires in the test with different duties. **(E)** The illustration of the SMA clutch. When heated, the SMA wire contracted and pulled the toothed slider to the left. Thus, the tooth slider was locked with the toothed screw. Then, the toothed screw started to rotate with the flexible inner pipe. When the SMA wire cooled down, an embedded compression spring pressed the toothed slider to its origin state. **(F)** Snapshots of the SMA clutch in its working progress. The SMA clutch was actuated in the 2.0 s. The power was cut off in the 12 s. The toothed slider unlocked the Toothed screw in the 14s. **(G)** The thermal map when the SMA clutch was working at room temperature (28 °C). **(H)** The thermal map when the SMA clutch was working at 39 °C.

Since the working temperature of the SMA wire was over 40 °C, safety was also a concern. We tested the maximum temperature of the SMA assembly in Fig. 3G. Since the SMA wire was isolated by a bundle of wires, and the heat was conducted and distributed by two layers of metal tube, the highest temperature was detected on the wire connected to the SMA wire directly. The highest temperature on the wires was 30.8 °C which was safe (Movie S5). The wires were further sealed in a plug. After being placed into the planar steerable pipe, the SMA clutch

should be safe in terms of the high temperature. To validate the performance in an environment with high temperatures (such as human organs), the SMA clutch was also placed on a thermostat table. In Fig. 3H, the environment was heated to nearly 40 °C. The SMA clutch was still working.

Comment 4:

Currently, many surgical intervention robots are made of flexible polymer materials, and adopt the pneumatically and hydraulically driven mechanisms. The proposed bioinspired robot avoids the commonly used driven mechanisms, which may increase the modeling complexity and response time. Please refer to more relevant literature and highlight the advantages of the proposed robot through experimental comparison.

Response:

Thank you for your suggestion. To the best of our knowledge, cable-driven continuum robots are still commonly used driven mechanisms for surgical intervention robots [1], [2]. Cable-driven continuum robots do exist complexity in modeling. The proposed robot consists of independently actuated bending modules. To actuate the robot, the operate decide the bending module to be activate and control few bending modules at a time. Thus, we think it's quite simple and user-friendly. The response time is scarified in our design to scale down the size of the robot. Benefiting from this design, the robot can operate in a remote site when constrained by tortuous paths. We further scaled down the diameter of the prototype to 0.8 cm.

This handheld robot is regarded as a low-cost, easy-to-use, and task-oriented approach for various kinds of medical scenarios. Such a prototype could be capable of traveling through the digestive tract for transluminal diagnosis and could support some complex operations such as the axial removal of foreign bodies demonstrated in the therapy experiment. In addition, it is conceived that it has the potential to become the carrier of surgical instruments in abdominal or pelvic endoscopic procedures, bypassing obstacles and reaching the deep lesion by multi-segmental motion coordination of the manipulator. We hope the prototype can be **assembled in situ** to meet surgical needs rather than manufactured to fit a specialized occasion.

Pneumatically and hydraulically driven mechanism presents a higher technical threshold to be scaled down and assembled in situ. The cable-driven bending modules actuated by a single flexible shaft are more feasible for us to realize the time-share driven design.

We demonstrate minimally actuated robots in the Discussion. However, we think there are technical difficulties for pneumatically or hydraulically driven mechanisms to be minimally actuated. To the best of our knowledge, their sections have to be driven by separated tubes such as the robot proposed by Zhang et al [3]. We added some relevant literature about minimally actuated designs.

----- Reference -----

- [1] Omisore, Olatunji Mumini, et al. "A Review on Flexible Robotic Systems for Minimally Invasive Surgery along with some of the technical and technological challenges hindering their prominence." IEEE Transactions on Systems, Man, and Cybernetics: Systems PP.99(2020).
- [2] P. E. Dupont, N. Simaan, H. Choset and C. Rucker, "Continuum Robots for Medical Interventions," in Proceedings of the IEEE, vol. 110, no. 7, pp. 847-870, July 2022, doi: 10.1109/JPROC.2022.3141338.
- [3] Boyu Zhang, Yingwei Fan, Penghui Yang, Tianle Cao, and Hongen Liao. Worm-Like Soft Robot for Complicated Tubular Environments. Soft Robotics. Jun 2019.399-413.<http://doi.org/10.1089/soro.2018.0088>

Several solutions, referred to as minimally actuated robots, were proposed in recent years to save the actuation packages while pursuing more DOFs. These ideas focus on matching the actuator with different motion sections along the linear shape of the robot and can be classified into three categories. In the first category, characterized by mobile actuators^{37,38}, the strategy is to alter the position of the motor to actuate different motion sections. The motion sections have to be chained (serial robot) with a track for locomotion of the actuator and it's low-efficiency to switch between motion sections. In the second category, characterized by a central distribution mechanism³⁹, the strategy is to arrange individual transmission mechanisms (usually cable-driven) for each motion section. The cables of transmission mechanisms are centralized to the robot base to be selected by clutches. These robots are cumbersome and show little potential to be compact with a mass of driving cables embedded inside the robot. The third category, based on tendon-driven continuum robots, adopts stiffening / shape-locking methods to change the actuatable sections⁴⁰. Multiple shape-locking methods were reviewed⁴¹, among which the SMA clutch was favored for fast response time and compact structure⁴². The clutches settled in the local section enable the robot to be distributed and modular assembled. However, these robots⁴³ are threaded with driving tendons/cables along the entire body, so they cannot be truly modular assembled and expandable. Furthermore, these sections are coupled with little independence. Essentially, the robot is switched between several configurations to perform different trajectories. Our snake-inspired design achieves a minimally actuated robot in a total modular assembly. The actuator generates a standard rotation input. Each motion module generates its motion form unlimited by other modules.

Comment 5:

Fig. 2(B) displays the joint shift in the bending progress. Why did the authors set the link lengths as 2.0 mm, 3.0mm and 4.5 mm? Fig. 4(B) shows the contraction rate of the SMA wire using PWM. Please detail the duty value selection in the following sentences:

“Three of the lines were under a constant duty PWM (20%, 50%, 80%). One was from full duty to 20% (100%, 70%, 40%, 20%, 20%...). The left one was from full duty to 50% (100%, 70%, 50%, 50%...).”

Response:

Thank you for your question.

-Why did the authors set the link lengths as 2.0 mm, 3.0mm and 4.5 mm?

For the **joint shift**, it is proposed to verify the approximation of a smooth arc based on discrete links. For the planar steerable pipe, the link length has to be smaller than 4.5 mm. Otherwise, the joint shift exceeds 30 μm when the joints are bent to 15 deg. We plotted the curves with 2.0 mm and 3.0 mm link lengths to show that the joint shift gets smaller with shorter links.

Fig. 2B The joint shift of interlocked tube pieces in the bending progress. The figure showed the joint shift changing with the deflection in different link lengths.

-Please detail the duty value selection in the following sentences:

“Three of the lines were under a constant duty PWM (20%, 50%, 80%). One was from full duty to 20% (100%, 70%, 40%, 20%, 20%...). The left one was from full duty to 50% (100%, 70%, 50%, 50%...).”

As for **the PWM**, we added several experiments to analyze the selection of duty. The goal is to shorten the response time to reach the maximum contraction and maintain the contracted state for the bending module to execute the bending motion. Thus, the actuation of the SMA clutch contains two stages.

In the first stage, we give a full-duty PWM input to expedite the contraction. The new experiment in Fig. 3B analyzes the selection of the voltage and pulse duration. The SMA wire is heated fast with a higher voltage input. Meanwhile, the pulse duration needs to be shortened or the SMA wire gets overheated with no further contraction.

In the second stage, we change the duty of the PWM input to maintain the temperature (also the displacement). A new experiment with results plotted in Fig. 3C and D analyzes the selection of duty. We changed the frequency of the PWM input to 100 Hz to reduce the fluctuation in the heating-cooling cycle. The proper duty is 4% when the SMA wire is placed on the homemade sliding table since the SMA wire kept the maximum contraction until the PWM was halted in the 12 s and the temperature was maintained at the low level (around 40 °C in Movie S3).

----- Manuscript Revised (Results / Page 4~5) -----

Pulse width modulation (PWM)³⁶ was introduced to reduce energy consumption. The voltage input was switched to a PWM input with five duties to maintain the contracted state. Among the five trials, the one with 4% duty maintained a relatively low temperature (40 °C) (Fig. 3B). The displacements of the five trials were recorded in Fig. 3D. The SMA wire reached the maximum displacement in a few milliseconds. Those trials that kept the temperature over 40 °C maintained the maximum contraction until the PWM input was stopped (Only the trial with 2% duty failed). The SMA with 4% duty experienced a shorter recovery time back to the original length. These tests suggested that we could adjust the voltage, pulse duration, and duty of the PWM to promote the contraction performance of the SMA wire.

To control the connection between the flexible inner pipe and the screw pair, the SMA wire was folded and hooked to a toothed slider as illustrated in Fig. 3E. The two ends of the SMA wire were pinned on a fixed ring and connected to two wires for power supply. The screw pair contained a toothed screw for the toothed slider to bite in. The tooth slider got released by a compression spring hidden between the toothed screw and the toothed slider. This movable structure was named the SMA clutch. Six more wires were embedded in the flexible inner

tube to activate subsequent SMA clutches. Two ends of each wire were connected to the pins of a female plug and a male plug respectively (Fig. S3 in the Supplementary materials).

To test the SMA clutch, a 4.0 V voltage input was plugged in that lasted 0.2 s, followed by the PWM input with 6% duty. The parameters were suggested by the results of the SMA wire test. As the result shown in Fig. 3F and Movie S4, the toothed slider locked the toothed screw in 0.1 s. During the power-on period (10 s), the toothed slider was firmly attached to the toothed screw. It took another 2 s to unloose the connection (Movie S4).

Since the working temperature of the SMA wire was over 40 °C, safety was also a concern. We tested the maximum temperature of the SMA assembly in Fig. 3G. Since the SMA wire was isolated by a bundle of wires, and the heat was conducted and distributed by two layers of metal tube, the highest temperature was detected on the wire connected to the SMA wire directly. The highest temperature on the wires was 30.8 °C which was safe (Movie S5). The wires were further sealed in a plug. After being placed into the planar steerable pipe, the SMA clutch should be safe in terms of the high temperature. To validate the performance in an environment with high temperatures (such as human organs), the SMA clutch was also placed on a thermostat table. In Fig. 3H, the environment was heated to nearly 40 °C. The SMA clutch was still working.

Fig. 3. Shape memory alloy works as a clutch for selective driving each BM. (A) Left: the setting for the SMA wire test; Right: the thermal map when the SMA wire was working. (B) The maximum displacement and temperature of the SMA wire with different durations of pulse input. The data contain three groups with voltage changing from 3.0 V to 5.0 V. (C) The temperature changes when heated by PWM inputs with different duties. The duties of the PWM inputs were 2%, 4%, 6%, 8%, and 10%. In each trial, the SMA wire was activated by a 4.0 V input at 2.0 s which lasted 0.2 s, then powered by the PWM inputs with different duties. The power was cut off in the 12 s and the cooling process was also recorded. (D) The displacement of the SMA wires in the test with different duties. (E) The illustration of the SMA clutch. When heated, the SMA wire contracted and pulled the toothed slider to the left. Thus, the tooth slider was locked with the toothed screw. Then, the toothed screw started to rotate with the flexible inner pipe. When the SMA wire cooled down, an embedded compression spring pressed the toothed slider to its origin state. (F) Snapshots of the SMA clutch in its working progress. The SMA clutch was actuated in the 2.0 s. The power was cut off in the 12 s. The toothed slider unloosed the Toothed

screw in the 14s. (G) The thermal map when the SMA clutch was working at room temperature (28 °C). (H) The thermal map when the SMA clutch was working at 39 °C.

Comment 6:

In the diagnosis and therapy experiments of the proposed robot, the author did not explain the application of three driven principles in the interaction process. For example, how did the “muscle” PSS, “vessel” FIS and “nerves” SMA cooperate to perceive the stomach environment and perform the desired motor behaviors?

Response:

Thank you for your question. The proposed robot is handheld by the operator. The operator regards the robot as the extension of his hand and select the bending module needs to steer towards the target.

When one bending module was selected, the SMA wire as the “nerves” connect to the appointed bending module is activated. The SMA clutch closed. Then, the bending direction and motor speed are set. The “vessel” starts to transmit the rotation. The activated “muscle” is actuated by this rotation. The selected bending module bends to the goal point.

Based on the judgement of the operator, each bending module may take several steps to reach ideal place. To simply illustrate this progress, we drew a timeline in two additional experiments shown in Fig. 4. In Fig. 4C, a waveform was drawn to indicate the “nerve” of which bending module was activated. The PWM input switched to a low duty to maintain the contraction state. Then the “vessel” works with the motor rotating. The activated “muscle” moved at the same time.

Fig. 4. The robot performs locomotion through a tortuous path and distal operation based on the “time-share driven” method. (A) The illustration of the time-share driven robot working in a remote place. The Motor base was placed behind the narrow entrance. After being lengthened by the flexible extension tube, the distal BMs were pushed through a tortuous path and reached a remote workspace. The distal section contains four BMs actuated by one flexible shaft inside of the flexible extension tube. **(B)** The assembling of two BMs in situ. Step 1: Connect the flexible inner pipes with an adapter. The GND pins were matched. Step 2: rotate the planar steerable pipe of the next BM to adjust its bending direction. The plug of the planar steerable pipe has eight phases. Two plugs have to be matched. Step 3: remove the adapter. Step 4: connect the planar steerable pipe directly. Their flexible inner pipes are connected at the same time. **(C)** The robot passes through a tortuous path. The tortuous path has two bends. Two BMs took a follow-the-leader motion illustrated by eight top-view snapshots marked 1–8. From snapshot 1–3, BM 2 turned left first to pass the first bend. BM 1 followed BM 2 and performed its left turn. From snapshot 3–5, BM 2 turned right twice to pass the second bend. From snapshot 5–7, BM 1 followed BM 2 and performed its two right turns. In snapshot 8, both BMs passed the two bends successfully. **(D)** The robot performs distal operations through a tortuous path. The distal section consists of three BMs marked 1, 2, and 3. BM 3 has a longer steerable section to obtain a larger bending range. A channel is attached to the tip of BM 3 to send a clamp. According to the position of the red target, the operator decided to actuate BM 2 and BM 3 to reach the target illustrated by six snapshots marked 1–6. From snapshots 1–3, BM 2 and 3 were actuated in sequence. From snapshot 4–5, the target was taken off by the clamp. From snapshot 5–6, BM 3 turned and removed the target.

Locomotion and operation based on a time-share driven method

The bioinspired time-share driven robot (Fig. S5 in the Supplementary materials) was assembled as shown in

Fig. 4A. The BMs can be assembled in situ as illustrated in Fig. 4B (Movie S6). The kinematic model of one BM and the time-share driven robot was explained in Supplementary materials (Fig. S6 and Fig. S7).

To exhibit the capability of the proposed robot to operate on a remote site heavily constrained by a narrow entrance and tortuous path, two experiments were conducted. The first was the locomotion through a tortuous path containing two bends (Fig. 4C). To pass through such a path, the follow-the-leader motion was adopted. The BM 2 in the front was the leader. BM 2 turned left and then moved ahead to pass the first bend. Then, it turned right and moved ahead to be ready for the second bend. At last, it turned right to pass the second bend. The BM 1 took the same motion as the leader and also successfully passed these two bends (Movie S7). This procedure was illustrated by the timeline drawn on the upper space of Fig. 4C. The second was the distal operation to remove a target (Fig. 4D). The robot contains three BMs and a clamping channel attached to the tip. The BMs were bent in the horizontal plane. The flexible extension tube went through a small opening and a tortuous path. The target was placed in a remote site. Based on the time-share driven method, two BMs were actuated to remove the target (Movie S8).

Comment 7:

Please give more information about the corrosion resistance and biosafety of the material in stomach acid, furthermore, also give more analysis of the experimental results of the clinical tests, rather than just stating the results only.

Response:

Thank you for your question. In the clinical therapy experiment, polytetrafluoroethylene was used to encapsulate the extension tube and polyurethane membranes were used to encapsulate the manipulator to ensure the biocompatibility and waterproofing of the robot. They are corrosion-resistant, and have a low friction coefficient, making them suitable as a coating for endoscopes that come into direct contact with biological tissue. For actual clinical application, the proposed robot needs to integrate more channels for suction, inflation, and flushing and be encapsulated by polytetrafluoroethylene as the currently used gastroscopy system, ensuring safe contact with human tissue and to meet the requirements of internal channel sterilization and external surface sterilization.

We modified the discussion and added a part of the analysis of clinical tests based on the operation performance of diagnosis and therapy.

----- Manuscript Revised (Results / Page 8) -----

For the therapy experiment (Fig. 6D), the robot was integrated with one more tool channel (2.4 mm, in Fig. 6E) for endoscopic clamp (2.0 mm) access. The inner and outer layers of the extension tube were composed of polytetrafluoroethylene, which is acid-resistant, self-lubricating, and harmless to biological tissues. Polyurethane membranes were used to encapsulate the BMs, which are biocompatible and flexible, making the manipulator waterproof. An ex vivo porcine stomach was properly secured on the operating table and slightly inflated (400 ml) by an air tube to enlarge the inner space (Fig. S9 in the Supplementary materials). There was a small opening in the fundus for tubular camera access to record the robot's movement in the gastral cavity. A fishbone (5 cm, in Fig. 6F), as a foreign body, was stuck in the lining of the stomach. The robot entered the gastral cavity via the esophagus and began examination in the same route of diagnosis experiment. After locating the fishbone, the robot adjusted to the planned removal direction at an angle of 30 ~ 45 deg from the axis of the fishbone, facing the sharp edge (Fig. 6G and H). The clamp was sent out through the tool channel to hold the bone, and the

manipulator bent to slowly lift it in the direction of bone insertion until it separated from the stomach wall (Fig. 6I), ensuring the fishbone tip was away from any tissue (Fig. 6H). Lastly, the bone was taken out by the robot withdrawn (Movie S6).

----- Manuscript Revised (Results / Page 9) -----

For clinical applications of the proposed robot, it needs to integrate more channels for suction, inflation, and flushing. Those channels can surround the robot and form a robotic endoscope system with a tube diameter of 1cm using the encapsulation of currently used gastroscopy. This gives the robot biocompatibility to ensure safe contact with human tissue and to meet the requirements of internal channel sterilization and external surface sterilization. The time-share driven mechanism provides an unprecedented way to bring the controllable DOFs and the flexibility of the continuum robot to the next level.

Reviewer #2

General Comment:

This manuscript introduces a handheld device that can selectively alter the bending motion of actuation modules using a single motor, through an SMA-based clutch mechanism. The mechanism and structure of the robot are very interesting and addresses the difficulties encountered in developing continuum robots, such as high degrees of freedom in a compact form. The authors also demonstrate its potential for use in medical interventions.

Comment 1:

However, the current form of the introduction needs to be rewritten as it presents general comments on robots, bioinspired robots, MIS, and other related topics in a poorly organized manner. In addition, the results section should focus on evaluating the robot's performance extensively, rather than solely introducing fabrication details related to its structure and control.

Response:

- However, the current form of the introduction needs to be rewritten as it presents general comments on robots, bioinspired robots, MIS, and other related topics in a poorly organized manner.

Thank you for your sincere advice. We have revised the introduction to elaborate on these topics more logically. For **bioinspired robots**, it remains unsettled to develop a robot with an enormous degree of freedom in a compact size as creatures represented by snakes. Inspired by the snake, we proposed a linear-shaped robot with several expandable bending modules actuated by only one motor. In this arrangement, the robot is compact and portable with an expandable degree of freedom. We are optimistic about its application in **minimal invasive flexible surgery**. Although current cable-driven continuum robots are compact and perform multi-degree-of-freedom movements, these movements are hard to operate on a remote surgical side via a tortuous path. Because the driven cables are off-center and influenced by the curved section along the path. The proposed robot is actuated by a flexible shaft. Adapting to the tortuous path, the flexible shaft can still actuate many bending modules to support multi-DoF operation. The handheld robot gets validated in this clinical scenario.

- In addition, the results section should focus on evaluating the robot's performance extensively, rather than solely introducing fabrication details related to its structure and control.

To **evaluate the robot's performance extensively**, we fabricated a new prototype and carried out more experiments on the new prototype. The Result section was rearranged into **four parts**. The first part validated the performance of a stainless-steel braided hose to **transmit precise rotation along a passively curved hose**. Based on this idea, the bending module was designed. The bending performance and the self-hold property were tested. The second part analyzes **the temperature and the contraction of the SMA**. Based on the results of these tests, a strategy for controlling the SMA clutch was proposed. The response and recovery time of the SMA clutch was analyzed. The temperature was also taken into consideration. The third part introduced the **in-situ assembling of the bending modules**. The experiments were divided into two aspects. First was **the movement along a tortuous path**. The second was **the operation on a remote site** accessed via a tortuous path. The last part remains a transluminal diagnosis experiment in a human stomach structure model and a transluminal therapy experiment in an excised swine stomach. The transluminal therapy experiment was accomplished by our new prototype with a diameter of 8 mm (the old one is 12 mm).

We hope these experiments provide a wide evaluation of the robot's performance and exhibit its key features.

.....

Based on this design, minimally actuated continuum robots are especially suitable for minimal invasive flexible surgery¹⁶ (MIFS) where the operational approaches are required to be less constrained by anatomical structures. This application necessitates the design of lightweight flexible manipulators with minimum footprint. In recent years, continuum robots¹⁷⁻²⁰ equipped with multiple DOFs by compacting more actuated cables/tendons have been proven to be an effective approach. However, these robots^{21,22} are actuated by a large number of motors in the robot base, making the robot system especially huge and expensive, and showing a sharp learning curve to surgeons. Furthermore, in terms of transluminal surgery for nearly non-invasive access and high-dexterous operation on distal surgical sites, the multi-DOF steerable manipulators extended by straight tube are unsuitable to establish an operation workspace through a tortuous path, while extension with passive flexible proximal section supports poor distal maneuvers over a long-distance transmission. Consequently, surgeons still face difficulties in effectively applying their techniques in the utilization of current devices.

Technically speaking, the stiffness of flexible devices is designed as a compromise between being stiff enough to enable distal propagation and operation and being compliant enough to adapt to the structural constraints along a tortuous path²³. To address this, it is required that the flexible robot reserves its shape and marches, stated as the follow-the-leader (FTL) motion²⁴. It was first proposed as an algorithm working on continuum robots with enormous DOFs. Lately, various FTL mechanisms²⁵ have been proposed in medical devices based on stiffening methods. To intrinsically follow the leader, two tubes take their turns to be the leader and get stiffened for the other to march forward²⁶. However, it remains unsettled to send the surgical instruments to the tip and enable its operations with sufficient DOFs. Our snake-inspired design fills up the last piece with a flexible rotation shaft. The flexible rotation shaft, metaphorized as a "vessel", can be extended to the very tip, and actuate endless BMs on the tip, thus providing as many DOFs as demanded.

Last but not least, the proposed prototype was elaborated as a handheld device to provide surgeons with direct enhancement of certain maneuvers, assumed to be susceptible by surgeons in clinical practice. The socketed BMs were installed on a handheld base housing one servo stepper motor. The robot consists of a chain of socketed bending modules (BM) installed on a handheld motor base. The bioinspired handheld time-share driven robot shows expansibility in a portable size over current continuum robots (Table S1 in the Supplementary materials). The applicability was proved in the clinical scenario: standard diagnosis of gastric ulcers and therapy of removing the foreign body (Fig. 1C). The time-share driven robot supports remote multi-DOF operations via a transluminal approach, promising to promote the robotic application in MIFS as a simple and effective solution.

Locomotion and operation based on a time-share driven method

The bioinspired time-share driven robot (Fig. S5 in the Supplementary materials) was assembled as shown in Fig. 4A. The BMs can be assembled in situ as illustrated in Fig. 4B (Movie S6). The kinematic model of one BM and the time-share driven robot was explained in Supplementary materials (Fig. S6 and Fig. S7).

To exhibit the capability of the proposed robot to operate on a remote site heavily constrained by a narrow entrance and tortuous path, two experiments were conducted. The first was the locomotion through a tortuous path containing two bends (Fig. 4C). To pass through such a path, the follow-the-leader motion was adopted. The BM 2 in the front was the leader. BM 2 turned left and then moved ahead to pass the first bend. Then, it turned right and moved ahead to be ready for the second bend. At last, it turned right to pass the second bend. The BM 1

took the same motion as the leader and also successfully passed these two bends (Movie S7). This procedure was illustrated by the timeline drawn on the upper space of Fig. 4C. The second was the distal operation to remove a target (Fig. 4D). The robot contains three BMs and a clamping channel attached to the tip. The BMs were bent in the horizontal plane. The flexible extension tube went through a small opening and a tortuous path. The target was placed in a remote site. Based on the time-share driven method, two BMs were actuated to remove the target (Movie S8).

Fig. 4. The robot performs locomotion through a tortuous path and distal operation based on the “time-share driven” method. (A) The illustration of the time-share driven robot working in a remote place. The Motor base was placed behind the narrow entrance. After being lengthened by the flexible extension tube, the distal BMs were pushed through a tortuous path and reached a remote workspace. The distal section contains four BMs actuated by one flexible shaft inside of the flexible extension tube. (B) The assembling of two BMs in situ. Step 1: Connect the flexible inner pipes with an adapter. The GND pins were matched. Step 2: rotate the planar steerable pipe of the next BM to adjust its bending direction. The plug of the planar steerable pipe has eight phases. Two plugs have to be matched. Step 3: remove the adapter. Step 4: connect the planar steerable pipe directly. Their flexible inner pipes are connected at the same time. (C) The robot passes through a tortuous path. The tortuous path has two bends. Two BMs took a follow-the-leader motion illustrated by eight top-view snapshots marked 1-8. From snapshot 1-3, BM 2 turned left first to pass the first bend. BM 1 followed BM 2 and performed its left turn. From snapshot 3-5, BM 2 turned right twice to pass the second bend. From snapshot 5-7, BM 1 followed BM 2 and performed its two right turns. In snapshot 8, both BMs passed the two bends successfully. (D) The robot performs distal operations

through a tortuous path. The distal section consists of three BMs marked 1, 2, and 3. BM 3 has a longer steerable section to obtain a larger bending range. A channel is attached to the tip of BM 3 to send a clamp. According to the position of the red target, the operator decided to actuate BM 2 and BM 3 to reach the target illustrated by six snapshots marked 1-6. From snapshots 1-3, BM 2 and 3 were actuated in sequence. From snapshot 4-5, the target was taken off by the clamp. From snapshot 5-6, BM 3 turned and removed the target.

Comment 2:

A rigorous literature review on continuum robots, specifically for minimally invasive surgery (MIS), is recommended. As the key contribution of this paper is believed to be the universal transmission and selective bending control, the authors should compare it with other approaches as outlined below.

- 1. A Novel Underactuated Continuum Robot With Shape Memory Alloy Clutches, Bishop et al. 2022. doi: 10.1109/TMECH.2022.3179812.*
- 2. Multiple Curvatures in a Tendon-Driven Continuum Robot Using a Novel Magnetic Locking Mechanism, Pogue et al. 2022. doi: 10.1109/IROS47612.2022.9982193.*
- 3. Continuum robots adopting follow-the-leader mechanism.*

Response:

Thank you for your sincere advice. The **follow-the-leader mechanism** was stated in the Introduction. In our opinion, the research on follow-the-leader was divided into two parts. One is the algorithm working on continuum robots with enormous degrees of freedom. Their robots present difficulty to be scaled down with many joints to be actuated. The other is the mechanical mechanism to inherently follow the leader. By stiffening one part of the robot, the rest part can march along the locked part (the leader). However, it's not expandable. Two parts take their turn to be actuated. This robot can pass through a tortuous path using this strategy. But it's not enough for tip operation. It also remains unsettled to send surgical instruments to the tip.

The underactuated continuum robots are introduced in the discussion. We regard these robots as **minimally actuated robots**, which aim at saving actuation packages while keeping many degrees of freedom. Their designs are reviewed and divided into three categories. These robots are viewed as a whole which switches between several configurations. Our robot is distinguished by its modular assembly. The bending modules are independently actuated. Thus, it's not limited to several configurations. In addition, we just fabricated a new prototype with a diameter of only 8 mm. To the best of our knowledge, no other minimally actuated robots can be scaled down to this size, and competent in medical applications.

----- Manuscript Revised (Introduction / Page 2) -----

Technically speaking, the stiffness of flexible devices is designed as a compromise between being stiff enough to enable distal propagation and operation and being compliant enough to adapt to the structural constraints along a tortuous path²³. To address this, it is required that the flexible robot reserves its shape and marches, stated as the follow-the-leader (FTL) motion²⁴. It was first proposed as an algorithm working on continuum robots with enormous DOFs. Lately, various FTL mechanisms²⁵ have been proposed in medical devices based on stiffening methods. To intrinsically follow the leader, two tubes take their turns to be the leader and get stiffened for the other to march forward²⁶. However, it remains unsettled to send the surgical instruments to the tip and enable its operations with sufficient DOFs. Our snake-inspired design fills up the last piece with a flexible rotation shaft. The flexible rotation shaft, metaphorized as a "vessel", can be extended to the very tip, and actuate endless BMs

on the tip, thus providing as many DOFs as demanded.

----- Manuscript Revised (Discussion / Page 8) -----

Several solutions, referred to as minimally actuated robots, were proposed in recent years to save the actuation packages while pursuing more DOFs. These ideas focus on matching the actuator with different motion sections along the linear shape of the robot and can be classified into three categories. In the first category, characterized by mobile actuators^{37,38}, the strategy is to alter the position of the motor to actuate different motion sections. The motion sections have to be chained (serial robot) with a track for locomotion of the actuator and it's low-efficiency to switch between motion sections. In the second category, characterized by a central distribution mechanism³⁹, the strategy is to arrange individual transmission mechanisms (usually cable-driven) for each motion section. The cables of transmission mechanisms are centralized to the robot base to be selected by clutches. These robots are cumbersome and show little potential to be compact with a mass of driving cables embedded inside the robot. The third category, based on tendon-driven continuum robots, adopts stiffening / shape-locking methods to change the actuatable sections⁴⁰. Multiple shape-locking methods were reviewed⁴¹, among which the SMA clutch was favored for fast response time and compact structure⁴². The clutches settled in the local section enable the robot to be distributed and modular assembled. However, these robots⁴³ are threaded with driving tendons/cables along the entire body, so they cannot be truly modular assembled and expandable. Furthermore, these sections are coupled with little independence. Essentially, the robot is switched between several configurations to perform different trajectories. Our snake-inspired design achieves a minimally actuated robot in a total modular assembly. The actuator generates a standard rotation input. Each motion module generates its motion form unlimited by other modules.

Comment 3:

Is there any concern regarding the total weight of the bending modules in operation? Could you please provide specifications for a single bending module? The authors have only mentioned the weight of the motor base.

Response:

-Is there any concern regarding the total weight of the bending modules in operation?

The proposed robot contains a length of extension tube to reach a remote surgical site via a tortuous path. The extension tube is flexible. Thus, the robot needs to be **supported by the environment**. After the flexible tube was supported by the surroundings, one bending module can support the weight of two to three bending modules as illustrated in Fig. 4A. It's sufficient for distal operation.

-Could you please provide specifications for a single bending module?

The weight of a single bending module is 18.2 g (the diameter is 12 mm). We fabricated an 8 mm bending module. The weight is only 8.6 g. We tested the 8 mm bending module in Fig. 2E and F.

Fig. R1. Two different bending modules.

----- Manuscript Revised (Results / Page 4) -----

The planar steerable pipe was fabricated with laser-profile technology (LPT)²⁷. A thin-walled stainless-steel tube was dismembered by LPT into a discrete-jointed tube containing a series of discrete interlocked segments. One lateral side of the tube was notched with columns of pits for driven tendons (Fig. S2 in the Supplementary materials). The flexible inner pipe contains a section of stainless-steel braided hose (Fig. S3 in the Supplementary materials). The braided layer could resist torsion, which enabled flexing within the planar steerable pipe while transmitting torsion along the pipe. The BM is assembled with one flexible inner pipe packed inside one planar steerable pipe (Fig. S4 in the Supplementary materials). The bending behavior of the BM (Movie. S2) is similar to other continuum designs²⁸⁻³¹. The bending range can be modified by adjusting the fixed point of the driven tendons on the screw pair. Thus, each BM has to be tested to record the bending range. The bending range was saved by the software to prevent the BM from exceeding its motion range and hitting gears. One bending module was tested in Fig. 2E. The bending speed was constrained by the motor (2 r/s rotation). The BM could hold a 50 g weight in Fig. 4F. One BM weighs only 8.6 g.

Comment 4:

How would the operator know the position of a threaded gear within the stroke of a screw, if it is not visible inside the human body? Is the threaded gear always centered when the system is at rest and without any bending? And how can one avoid hitting the gear at either end of the screw?"

Response:

- How would the operator know the position of a threaded gear within the stroke of a screw, if it is not visible inside the human body?

The bending range of one bending module is determined in the fabrication stage. Two bending modules may not have the same bending range. Thus, we tested each bending module (explained in Fig. 2E in the manuscript) and **recorded its motion range**.

- Is the threaded gear always centered when the system is at rest and without any bending?

Before inserting it inside a cavity, each bending module is adjusted to a straight state. It may not be the center, because we need some bending module steer more toward one side.

- And how can one avoid hitting the gear at either end of the screw?

Then, for each bending module, the number of turns (flexible inner pipe) is recorded by the program. In this way, we can avoid hitting the gears. By the way, the bending modules are not designed for long-term use and can be assembled in situ. If one bending module does not meet our needs in terms of motion range, we can replace it.

----- Manuscript Revised (Results / Page 4) -----

Fig. 2. Design and performance of the BM actuated by a flexible shaft. (E) One BM performed steering to both sides. This BM could bend 38 deg toward the left side and 30 deg toward the right side. The bending angle was plotted in a blue line. The speed was plotted in the red line. The speed reached 10 deg per second.

Specific comment 1:

Title: In the current version of the title, "expandable dexterity" is used, but the robot described in the manuscript is actually designed for expandable degrees of freedom (DOF) rather than dexterity. The manuscript does not evaluate the dexterity of the robot. To show dexterity with this robot, the robot should be able to maneuver surgical tools with precision and flexibility.

Response:

Thank you for your suggestion. We revised the expression of "dexterity". The current prototype is operated by surgeons. It necessitates the judgment of the operator to finish certain procedures. Thus, we emphasize the expandable degrees of freedom (DOF) as suggested.

----- Manuscript Revised (Title) -----

Title: Bioinspired handheld time-share driven robot with expandable DoFs

Specific comment 2:

Line 38-39: The current form of the robot seems to have less controllability and precision?

Response:

Thank you for your comment. The idea of the time-share driven design is that the operator images lots of joints equipped to his hand. He can actuate different bending modules to access the surgical site. Usually, only one bending module is moved at a time. Thus, the controllability of one bending module is quite well as shown in movie S2.

In the current stage the robot was operated in handheld mode. The controllability and precision were not problems and validated in Fig. 4D. Movie S8 demonstrated how the robot removed a target in a remote site.

----- Manuscript Revised (Results / Page 6) -----

Fig. 4. The robot performs locomotion through a tortuous path and distal operation based on the “time-share driven” method. (D) The robot performs distal operations through a tortuous path. The distal section consists of three BMs marked 1, 2, and 3. BM 3 has a longer steerable section to obtain a larger bending range. A channel is attached to the tip of BM 3 to send a clamp. According to the position of the red target, the operator decided to actuate BM 2 and BM 3 to reach the target illustrated by six snapshots marked 1-6. From snapshots 1-3, BM 2 and 3 were actuated in sequence. From snapshot 4-5, the target was taken off by the clamp. From snapshot 5-6, BM 3 turned and removed the target.

Specific comment 3:

Line 54-55: The surgeon’s learning curve may substantially vary the types of user interfaces used for control.

Response:

Thank you for your comments. Indeed, the clinical application and popularization of any surgical instrument will face the problem of technical threshold and learning curve. The controlling pattern of the proposed robot is user-friendly: the operator can control the movement of each bending module through the knob on the handheld base, and can also control the rotation of the robot freely. This is very similar to the current controlling pattern of gastroscopy. Combined with the implementation and operation experience, we have reason to believe that the learning curve of this instrument and procedure is gentle.

Specific comment 4:

Line 68: State Keeping.

Is there any potential issue in “state keeping” with the backlash of the lead screw and mechanical clearance in manufacturing and fabrication process or structural stiffness? Any clearance would be accumulated along the serial connection and affect the end pose of the robot.

Response:

Thank you for your comment.

The **backlash and clearance**, especially along the axial direction, would change the displacement of the driven tendons and thus affect the end pose of the robot. These factors need to be looked at separately. The backlash of the lead screw is very tiny ($<3\mu\text{m}$). The unified fine thread is machined by computer numerical control (CNC). The clearance exists between the screw pair and the straight tube of the planar steerable pipe because the screw pair needs to rotate in the straight tube. To control this clearance along the axial direction in the fabrication process, we adjust the two parts which fix the screw pair in the straight tube. The clearance can be adjusted to 0.2 mm.

The structural stiffness needs to be considered separately. The state-keeping we mentioned is about the screw pair. The screw pair cannot be driven by external force. The bending module contains a section of continuum

structure. To pass through tortuous path and protect the surroundings, it is elastic. Some continuum robots based on discrete joints contain wires to lock all the joints to keep their shape, referred to as “state keeping”. Our robot does not use this technique.

Indeed, both the clearance and the elastic deformation would accumulate and affect the end pose of the robot. The robot accomplished the task based on a relative location without sensor feedback. In the future, we plan to embed a fiber optic displacement sensor in the robot to measure the tip position.

Specific comment 5:

Fig 2B: what is joint shift of interlocked tube pieces? It was hard to understand why 4.5 mm value was adopted in the current design through the manuscript.

Response:

Thank you for your comment. The joint shift is proposed to eliminate the bias in the approximation of the arc using discrete links. If the joints of discrete link have no clearance, the tip position of the discrete-link robot is not the same as that of the continuum robot. We fixed the arc length of the discrete-link structure with tendons. Then, the length of the discrete link is regard as a variable to fit the arc of tendons. The length change of the discrete link should not exceed the clearance of laser-cutting, or the joint shift will not be realized.

The link length cannot be too short or the discrete links are too weak. The 4.5 mm is the maximum link length that the clearance of laser-cutting can tolerate. So, we adopted 4.5 mm link length.

Fig. R2 Joint shift

For the flexible inner pipe and extension tubes, we replaced the laser-cut tube with a **stainless-steel braided hose**. The new hose is a continuum structure without discrete joints. The hose has no joint clearance and can resist torsion as analyzed in Fig. 2. It has a much better performance than the old design.

-----Supplementary materials (Fig. S1) -----

2. Details of the stainless-steel braided hose.

Fig. S1. Details of the stainless-steel braided hose. (A) The illustration of the stainless-steel braided hose. **(B)** The extension tubes. **(C)** The flexible inner pipe.

The stainless-steel braided hose contains three layers (Fig. S1A). The inner layer provides lubrication for inserting other shafts. The middle layer is a braided SS304 net. This layer resists torque and prevents buckling. The inner layer and outer layer are made of Polytetrafluoroethylene (PTFE), which is characterized by cold and heat resistance (180-260°C), acid and alkali resistance, resistance to various organic solvents, and its friction coefficient is very low, making it suitable as a coating for endoscopes that come into direct contact with human tissue.

Specific comment 6:

Line 88-89: This tube could be bent to a minimum radius of 17.2 mm and was much more rigid to resist out-of-plane forces than the continuum tube.

How was a minimum radius of 17.2 mm derived? Is it from FEA?

It would be helpful to show the stiffness of the structure for each direction or DOF using FEA simulation (e.g. torsional stiffness instead of lifting weight).

Response:

- How was a minimum radius of 17.2 mm derived? Is it from FEA?

Thank you for your comment. The minimum radius is calculated based on the maximum bending angle of each discrete joint, not from FEA.

- It would be helpful to show the stiffness of the structure for each direction or DOF using FEA simulation (e.g. torsional stiffness instead of lifting weight).

The structure is dismembered into discrete joints as a bendable tube. The bending motion is formed by the accumulation of small rotations, not the deformation of the stainless-steel tube, so it won't work out to calculate the stiffness of the structure. The stiffness is provided by the nitinol tendons threaded on the structure.

The new flexible inner pipe and extension tubes are made of a stainless-steel braided hose that can transmit torque. We tested the torsional stiffness in Fig. 2.

-----Manuscript Revised (Results / Page 3)-----

Fig. 2. Design and performance of the BM actuated by a flexible shaft. (D) The torsional rigidity test of the flexible extension tube. The inner hose was rotated from the free side end and the torque was recorded from the fixed side. The outer hose was straight and bent to 90 deg and 180 deg in three trials. The data were fit with straight lines drawn with dashed lines. The slopes of the lines were marked as torsional rigidity.

Specific comment 7:

Line 92: what is S14?

Response:

Thank you for your comment. S14 is an equation in supplementary materials.

$$d' - L_{link} = \left(\frac{\left(\frac{4 \sin\left(\frac{\theta}{2}\right)}{\theta} \right)}{\text{norm} \left(\begin{bmatrix} \frac{s_2}{2} \\ -s_1 \left(1 + \frac{c_2}{2} \right) \\ c_1 \left(1 + \frac{c_2}{2} \right) + \frac{1}{2} \end{bmatrix} \right)} - 1 \right) \cdot L_{link} \leq 30\mu m \quad (S14)$$

It's proposed to calculate the joint shift. We removed this part in the revised manuscript because this problem does not exist on the stainless-steel braided hose used in our new prototype.

Specific comment 8:

Line 95: making it suitable for long-distance transmission,

I would concern non-uniform torque transmission at the end. Please evaluate such long distance transmission in terms of torque loss? For example, depending on number of BM modules, external load and bending angles.

Response:

Thank you for your suggestion. We analyzed the long-distance transmission of the extension tubes in the revised manuscript (please check Fig. 2). The rotation loss was tested by an optical tracker. The external torque was also measured. The extension tubes are quite long (548 mm), while the flexible inner pipe contains only 33 mm flexible

tubes. So, the experiment was only conducted on the extension tubes.

Fig. 2. Design and performance of the BM actuated by a flexible shaft. (A) The time-share driven robot prototype is attached with only the flexible extension tube. (B) The components of the extension tube. The extension tube contains two stainless-steel braided hoses. A 3.9 mm hose rotates inside of a 6.1 mm one. All the subsequent BMs attached to the tube are actuated by the relative rotation between the two hoses. The control signals are transmitted to each BM via the wires embedded inside the inner hose. Two ends of the inner hose are a female plug and a male plug. Each contains 6 pins. One pin is marked as the common ground wire (GND). The outer hose ends with a pair of eight-jaw plugs. They have eight phases of connection to adjust the rotation plane of the subsequent BM. (C) The rotation test of the extension tube. The outer hose was bent from 0 deg to 180 deg. The rotation was plotted from -180 deg to 180 deg. A small figure to zoom in on the data between -6 deg and 6 deg was attached to each figure in below. In each trial, the motor was programmed to repeat an action cycle, in which it rotated one round and rested for 4 s. The inner hose was actuated by the motor from the base, while the rotation of the free end was recorded by the optical marker. (D) The torsional rigidity test of the flexible extension tube. The inner hose was rotated from the free side end and the torque was recorded from the fixed side. The outer hose was straight and bent to 90 deg and 180 deg in three trials. The data were fit with straight lines drawn with dashed lines. The slopes of the lines were marked as torsional rigidity. (E) One BM performed steering to both sides. This BM could bend 38 deg toward the left side and 30 deg toward the right side. The bending angle was plotted in a blue line. The speed was plotted in the red line. The speed reached 10 deg per second. (F) One BM held a 50 g still weight.

Specific comment 9:

Line 102: Please describe the specification of the super-elastic tendon used in the supplementary.

Was any pre-load applied on fixed tendon?

Response:

Thank you for your comments. The super-elastic tendon is a Ni-Ti alloy containing 55% Ni and 45% Ti. There was no pre-load applied on the fixed tendons. We used the fixed tendons to constrain the distance between disks

on the planar steerable pipe. The length of the fixed tendons has to match the planar steerable pipe. Thus, we cannot apply a pre-load on the fixed tendons or the length would change.

In our new prototype, the fixed tendons were removed. In the new design of the bending module, the driven tendons are directly threaded on notches crushed on the planar steerable pipe.

----- Supplementary materials (Fig. S2)-----

3. A schematic view of the planar steerable pipe

Fig. S2. A schematic view of the planar steerable pipe.

The planar steerable pipe was divided into a rigid section and a steerable section. The steerable section was fabricated with laser-profile technology (LPT). The stamped pits to house the elastic driven tendons were presented in the top view of the pipe. The elastic driven tendon is a kind of Ni-Ti alloy containing 55% Ni and 45% Ti. The rigid section contained a slotted pipe. The sliding nut was actuated by the toothed screw and moved along the slot. Two ends of the planar steerable pipe were closed with a female plug and a male plug separately. The two plugs had eight teeth. The connection of the plugs had eight phases.

Specific comment 10:

Line 69-70: The system assumes no length change in FIS because it lies along the neutral axis of bending. Does the curve of FIS always lie on the neutral axis? If not, it may undergo length change.

Response:

Thank you for your comments. Indeed, if the planar steerable pipe was bent to a larger angle, the flexible inner pipe got bent to press the inner wall of the planar steerable pipe and deviated from the neutral axis. A small length change occurred. However, the two ends of the flexible inner pipe were constrained to stay at the neutral axis of the planar steerable pipe. The length change did not affect the rotations of the two ends directly. Thus, it has little influence on the rotation transition.

The length change is not to be neglected on long tubes such as the extension tubes (548 mm). The length change on the inner hose of the extension tubes is quite large when bent over 90 deg. The plugs on the inner hose may be detached if a large length change happens. Thus, we designed a **floating connector** on the motor base. It can

change the position of the outer hose on the motor base and thus compensate for the length change of the inner hose adaptively. This is realized by three circular distributed rods with compression springs sliding on them.

----- Supplementary materials (Fig. S5) -----

6. A schematic view of the time-share driven robot

Fig. S5. A schematic view of the time-share driven robot.

The time-share driven robot can be held by the operator or fixed on a robotic manipulator. The total weight of the control box and the motor base was 480 g. Inside the control box were the STM32 controller board, the Voltage regulator module, and the multi-channel MOSFET module with multi-channel PWM input from the STM32 controller. The motor base was attached to the front side of the control box. The extension tube was attached to the motor base with the inner hose connected to the Output shaft. The outer hose was plugged into the Floating connector. The Floating connector contained three compression springs to adapt the relative length change of the outer hose with respect to the inner hose.

Specific comment 11:

Line 113~: The section Third principle - selective response includes,

The amount of detail provided in this section is excessive and would be more appropriate for inclusion in the supplementary materials, such as the instructions on how to wind SMA and control its duty. Therefore, I suggest that you focus instead on describing how the contraction of the expansible tubular structure is utilized to hold the screw and facilitate rotation with FIS selectively.

Response:

Thank you for your suggestion. We removed the force test and the SMA wire selection in the revised manuscript.

New tests were added to analyze the displacement and temperature of the SMA wire which directly determine the performance of the SMA clutch.

Also, we found that it was extremely difficult to analyze the contraction of the expansible tubular structure when it was already placed inside the hollow screw pair. The contact state was also invisible. It was troublesome when the deformation of the tubular structure was insufficient and the structure slipped inside of the screw pair. To solve this problem, we designed a different SMA clutch. The SMA wire dragged a toothed slider to bite a toothed screw to actuate the screw pair. A compression spring was placed between the toothed screw and the toothed slider to help the recovery of the SMA clutch.

In the updated design, the slipping is avoided because the locking direction is vertical to the rotation motion. The toothed slicer cannot be cast off by the torque. Additionally, the state can be directly observed from outside. As shown in Fig. 3F in the revised manuscript, we evaluated the locking and recovery time of the new SMA clutch.

-----Manuscript Revised (Results / Page 5)-----

Fig. 3. Shape memory alloy works as a clutch for selective driving each BM. (A) Left: the setting for the SMA wire test; Right: the thermal map when the SMA wire was working. (B) The maximum displacement and temperature of the SMA wire with different durations of pulse input. The data contain three groups with voltage changing from 3.0 V to 5.0 V. (C) The temperature changes when heated by PWM inputs with different duties. The duties of the PWM inputs were 2%, 4%, 6%, 8%, and 10%. In each trial, the SMA wire was activated by a 4.0 V input at 2.0 s which lasted 0.2 s, then powered by the PWM inputs with different duties. The power was cut off in the 12 s and the cooling process was also recorded. (D) The displacement of the SMA wires in the test with different duties. (E) The illustration of the SMA clutch. When heated, the SMA wire contracted and pulled the toothed slider to the left. Thus, the tooth slider was locked with the toothed screw. Then, the toothed screw started to rotate with the flexible inner pipe. When the SMA wire cooled down, an embedded compression spring pressed the toothed slider to its origin state. (F) Snapshots of the SMA clutch in its working progress. The SMA clutch was actuated in the 2.0 s. The power was cut off in the 12 s. The toothed slider unloosed the Toothed screw in the 14s. (G) The thermal map when the SMA clutch was working at room temperature (28 °C). (H) The thermal map when the SMA clutch was working at 39 °C.

Specific comment 12:

Line 118-119: The 100 μm SMA wire failed to drive a payload of 1000 g weight.

Why was 1000g payload tested? Is it due to the stiffness of the expansile tubular structure?

Response:

Thank you for your comments. Yes, the expansile tubular structure has to be compressed with a large force to be expanded sufficiently, so that the screw pair can be actuated by the friction. The weight was determined by experience. We changed the design of the SMA clutch. As responded in specific comment 11, the new locking mechanism is more visualized, reliable, and durable.

Specific comment 13:

Line 122: As the PWM test depicted in Fig. 4B, the best control strategy is to start with a full-duty PWM to shorten the response time, then drop the duty to a low level (50%) to avoid over-heating.

So, what was the temperature on the SMA element with each control method?

The contraction and recovery (cooling) behavior would be totally different in the human organs.

Response:

Thank you for your comments. We appreciate the suggestion that temperature needs to be analyzed in the application of the SMA clutch. We designed experiments to measure the temperature during the working state of the SMA wire. More videos were included in the Supplementary materials (Movie S3, S4, and S5). The duty of the PWM is determined accordingly. We also tested the performance of the SMA clutch on a heating table with a temperature a little higher than human organs.

-----Manuscript Revised (Results / Page 4~5)-----

Pulse width modulation (PWM)³⁶ was introduced to reduce energy consumption. The voltage input was switched to a PWM input with five duties to maintain the contracted state. Among the five trials, the one with 4% duty maintained a relatively low temperature (40 °C) (Fig. 3B). The displacements of the five trials were recorded in Fig. 3D. The SMA wire reached the maximum displacement in a few milliseconds. Those trials that kept the temperature over 40 °C maintained the maximum contraction until the PWM input was stopped (Only the trial with 2% duty failed). The SMA with 4% duty experienced a shorter recovery time back to the original length. These tests suggested that we could adjust the voltage, pulse duration, and duty of the PWM to promote the contraction performance of the SMA wire.

To control the connection between the flexible inner pipe and the screw pair, the SMA wire was folded and hooked to a toothed slider as illustrated in Fig. 3E. The two ends of the SMA wire were pinned on a fixed ring and connected to two wires for power supply. The screw pair contained a toothed screw for the toothed slider to bite in. The tooth slider got released by a compression spring hidden between the toothed screw and the toothed slider. This movable structure was named the SMA clutch. Six more wires were embedded in the flexible inner tube to activate subsequent SMA clutches. Two ends of each wire were connected to the pins of a female plug and a male plug respectively (Fig. S3 in the Supplementary materials).

To test the SMA clutch, a 4.0 V voltage input was plugged in that lasted 0.2 s, followed by the PWM input with 6% duty. The parameters were suggested by the results of the SMA wire test. As the result shown in Fig. 3F and Movie S4, the toothed slider locked the toothed screw in 0.1 s. During the power-on period (10 s), the toothed slider was firmly attached to the toothed screw. It took another 2 s to unloose the connection (Movie S4).

Since the working temperature of the SMA wire was over 40 °C, safety was also a concern. We tested the maximum temperature of the SMA assembly in Fig. 3G. Since the SMA wire was isolated by a bundle of wires, and the heat was conducted and distributed by two layers of metal tube, the highest temperature was detected on the wire connected to the SMA wire directly. The highest temperature on the wires was 30.8 °C which was safe (Movie S5). The wires were further sealed in a plug. After being placed into the planar steerable pipe, the SMA clutch should be safe in terms of the high temperature. To validate the performance in an environment with high temperatures (such as human organs), the SMA clutch was also placed on a thermostat table. In Fig. 3H, the environment was heated to nearly 40 °C. The SMA clutch was still working.

Fig. 3. Shape memory alloy works as a clutch for selective driving each BM. (A) Left: the setting for the SMA wire test; Right: the thermal map when the SMA wire was working. (B) The maximum displacement and temperature of the SMA wire with different durations of pulse input. The data contain three groups with voltage changing from 3.0 V to 5.0 V. (C) The temperature changes when heated by PWM inputs with different duties. The duties of the PWM inputs were 2%, 4%, 6%, 8%, and 10%. In each trial, the SMA wire was activated by a 4.0 V input at 2.0 s which lasted 0.2 s, then powered by the PWM inputs with different duties. The power was cut off in the 12 s and the cooling process was also recorded. (D) The displacement of the SMA wires in the test with different duties. (E) The illustration of the SMA clutch. When heated, the SMA wire contracted and pulled the toothed slider to the left. Thus, the tooth slider was locked with the toothed screw. Then, the toothed screw started to rotate with the flexible inner pipe. When the SMA wire cooled down, an embedded compression spring pressed the toothed slider to its origin state. (F) Snapshots of the SMA clutch in its working progress. The SMA clutch was actuated in the 2.0 s. The power was cut off in the 12 s. The toothed slider unlocked the Toothed screw in the 14s. (G) The thermal map when the SMA clutch was working at room temperature (28 °C). (H) The thermal map when the SMA clutch was working at 39 °C.

Specific comment 14:

Line 139-140: The motor rotated at a speed of 1 r/s. A 6 s PWM output was sent to BM 1, then BM 2, and ended up with an 8 s PWM to motivate both BMs (Movie S3).

Why was 8 s taken for driving the two modules, was it due to loss?

Response:

Thank you for your concern. In the test presented in Movie S3, we were not bothered by the loss. We wanted to make sure two bending modules not exceeding they motion ranges. 6 s PWM output was first selected as a test for two bending modules. We then found out it was feasible to give a longer PWM output. When two bending modules were moving at the same time, we set the time to 8 s.

We were not sure because the motor kept running at this mode. It had to be test when the SMA clutch locked and the screw pair rotated with the motor.

Specific comment 15:

Line 148~ In Diagnosis and therapy experiments of the time-share driven robot section and Movie S5 & S6, one or two BM modules are only shown. It would be better to show how the robot can pass through narrow openings and along tortuous paths (e.g. how the multi-segment robot works with time-share control). Fish bone removal in Movie S6 does not demonstrate the advantage of this robot. It seems to be just operation of an endoscopic clamp.?

Response:

Thank you for your suggestion. New tests were added in Fig. 4 to show the robot passing through small openings and along tortuous paths.

A new illustration of the clinical scenario including a diagnosis experiment on a human stomach model and a therapy experiment on an ex-vivo porcine stomach was added in Fig. 6. Two detailed videos (Movie S9 and S10) of the clinical scenario were added to the supplementary material.

Current gastroscopy systems have problems dealing with foreign objects in the esophagus or stomach, such as stuck fish bones. After the surgeon clamps the fish bone, it can only be pulled out along the axial direction of the gastroscopy, which can easily cause the stomach lining to be cut by the sharp edge of the bone. The proposed robot can flexibly adjust the tip to remove the fish bone along the inserted direction, minimizing the damage to the tissue, as shown in Fig. R3.

Fig. R3. Different ways to remove foreign bodies using a gastroscope and the proposed robot.

Fig. 4. The robot performs locomotion through a tortuous path and distal operation based on the “time-share driven” method. (C) The robot passes through a tortuous path. The tortuous path has two bends. Two BMs took a follow-the-leader motion illustrated by eight top-view snapshots marked 1–8. From snapshot 1–3, BM 2 turned left first to pass the first bend. BM 1 followed BM 2 and performed its left turn. From snapshot 3–5, BM 2 turned right twice to pass the second bend. From snapshot 5–7, BM 1 followed BM 2 and performed its two right turns. In snapshot 8, both BMs passed the two bends successfully.

Specific comment 16:

Supplementary Materials

1. Comparison of this work with continuum robot designs.

This prototype also entails a large controller back of the motor base. So, it may be unfair the soft robot air/water pumps are heavy.

Response:

Thank you for your comments. The controller can be scaled down. In the new prototype, we attached the motor base to the control box. All the electronic components are packed in one robot base. The total weight is 494.6 g. We assume that the robot is still portable and can be handheld.

Reviewer #3

General Comment:

I find the robot very interesting. However, from the description of the robot in the paper it is very hard for me to understand how exactly it works and what the exact design requirements were. That is a pity because the robot seems quite novel to me. The working principle of the robot requires a much more clear explanation at a functional level. Specifically:

Comment 1:

Introducing many abbreviations in lines 43-51 makes the paper very hard to read and to understand. Use less abbreviations in the paper to make it easier to read.

Response:

Thank you for your suggestion. We removed the abbreviations of the planar steerable pipe, flexible inner pipe, and the SMA clutch. We felt sorry that these abbreviations bothered your reading.

-----Manuscript Revised (Introduction / Page 1)-----

Snakes are unique vertebrates holding enormous flexibility. Their spinal cord travels along the vertebral canal, branching out spinal nerves at every level to connect muscle fibers, while the major distributing arteries carry blood pumped by the heart, branching into successively smaller vessels, ending as capillaries with nutrient supply to muscles. The stimulated muscles actuate multiple segmental spines, supporting multi-DOF motion. Inspired by snakes, we reported a novel time-share driven robot (Fig. 1A) with expandable DOFs by employing one motor to actuate multiple serially connected bending modules (BMs). Each BM (Fig. 1B) contains a planar steerable pipe, a flexible inner pipe, and an interbedded shape memory alloy (SMA) clutch. The planar steerable pipe is deflected by tendons as the “vertebra” is dragged by “muscles”. With voltage signal input, the SMA clutch activates the planar steerable pipe as the “nerve”. The flexible inner pipe rotates to actuate the stimulated BMs as the “vessel”. Similar to the case that blood vessels are perfused by a single heart, all the flexible inner pipes are linked and powered by the only motor. The robot is economical, compact, and portable by sharing one high-performance motor. Importantly, unbound to the motor, BMs can be tailored and reconfigured in task-oriented assembly.

Comment 2:

Lines 56-57 “the controllable degrees-of-freedom supplied by remotely actuated tendons are consumed quickly in the wandering of obstructive tissues”. No idea what you mean with this “wandering” – further explanation needed.

Response:

Thank you for your comment. We modified the corresponding part in the introduction section concerning minimal invasive flexible surgery. We used the word “wandering” to describe the non-invasive and tortuous access to a remote surgical site. For example, if the surgical site is in the stomach or even far such as in the small intestine, the robot needs a long and flexible section to extend for traveling along natural orifices. We removed the word since it confused you. The idea was rephrased.

Based on this design, minimally actuated continuum robots are especially suitable for minimal invasive flexible surgery¹⁶ (MIFS) where the operational approaches are required to be less constrained by anatomical structures. This application necessitates the design of lightweight flexible manipulators with minimum footprint. In recent years, continuum robots¹⁷⁻²⁰ equipped with multiple DOFs by compacting more actuated cables/tendons have been proven to be an effective approach. However, these robots^{21,22} are actuated by a large number of motors in the robot base, making the robot system especially huge and expensive, and showing a sharp learning curve to surgeons. Furthermore, in terms of transluminal surgery for nearly non-invasive access and high-dexterous operation on distal surgical sites, the multi-DOF steerable manipulators extended by straight tube are unsuitable to establish an operation workspace through a tortuous path, while extension with passive flexible proximal section supports poor distal maneuvers over a long-distance transmission. Consequently, surgeons still face difficulties in effectively applying their techniques in the utilization of current devices.

Comment 3:

Line 58 “surgeons frequently find a target unreachable”. To my knowledge such robots are not yet used by surgeons; they are still in the research phase. So what exactly do you mean here?

Response:

Thank you for your comment. Indeed, although lots of continuum robots are proposed to get access to surgical sites in a flexible manner, they are still in the research phase. Here, we meant the several surgical instruments that were widely applied. Currently, the gastroscope is a flexible electronic endoscope with three degrees of freedom of movement at the end of the structure. The gastroscope can pass through the tortuous path, adapting to the human body cavity without causing damage. However, when such a completely flexible endoscope moves in a larger cavity, such as the stomach, it often occurs that the operator can capture the target lesion in the endoscopic field of view but cannot adjust the approach angle flexibly for exerting operation due to the lack of external support. What’s more, gastroscopy has problems dealing with foreign objects in the esophagus or stomach, such as stuck fish bones. After the surgeon clamps the fish bone, it can only be pulled out along the axial direction of the gastroscopy, which can easily cause the tissue to be cut by the sharp edge of the bone. The proposed robot can flexibly adjust the tip to remove the fish bone along the inserted direction, minimizing the damage to the tissue, as shown in Fig. R3.

Fig. R3. Different ways to remove foreign bodies using a gastroscope and the proposed robot.

Comment 4:

Lines 60-62: can you show photos of the robot used in this experiment, can you give more information about this experiment, with what continuum robot did you exactly compare it? Figure 1 shows only drawings of a robot; it does not show a real robot. So I do not understand how you tested the applicability in a clinical scenario, lines 62-63. Or do you refer in this section to another experiment explained in an already published paper? Add then a reference to the paper where I can find the details of this experiment.

Response:

Thank you for your comment.

Current continuum robots for medical intervention are reviewed in two references listed below.

[1] Omisore, Olatunji Mumini, et al. "A Review on Flexible Robotic Systems for Minimally Invasive Surgery along with some of the technical and technological challenges hindering their prominence." *IEEE Transactions on Systems, Man, and Cybernetics: Systems* PP.99(2020).

[2] P. E. Dupont, N. Simaan, H. Choset and C. Rucker, "Continuum Robots for Medical Interventions," in *Proceedings of the IEEE*, vol. 110, no. 7, pp. 847-870, July 2022, doi: 10.1109/JPROC.2022.3141338.

Although few robots were used clinically, many kinds of flexible robots were proposed to enhance minimally invasive interventions on internal organs located in confined areas of the human body. These surgical devices are designed to navigate anatomical pathways via single-port access, such as natural orifices or minimal incisions and intraluminal interventions.

These robots were compared in the design aspects. This is why we present the drawings of a robot to catch the basic idea of these robots. Wireless actuation (such as magnetically actuated robots) is not discussed in the comparison since they can only navigate through an anatomical pathway. The complex operation of the distal end is hard to perform in the current stage.

Current flexible robots represented by Auris Monarch robotic bronchoscope illustrate the basic form of robots suitable for minimally invasive surgery. Their manipulators are divided into a long and passively flexible section and a controllable distal section. Limited to the long and tortuous anatomical pathways, it presents difficulties in actuating the distal parts. Our work is distinguished by the only flexible rotary shaft to actuate a distal part containing several bending modules. The rotation is the only motion form that can transmit along a curvilinear path and is not affected by the bent shape. Other tendon-driven robots have to compensate for the length change in the flexible section.

The proposed robot was not specialized for gastric therapy. We selected this scenario because it exhibits the features of our robot. We pretend to further scale down the size of the proposed robot to get access to more anatomical pathways. A new prototype was fabricated with a diameter of 8 mm.

Fig. R1. Two different bending modules.

Comment 5:

End of introduction: mention what exactly the goal is of the research presented in the paper. What are the design requirements of the robot in terms of overall diameter, length, force that it can exert on the environment, etc.?

Response:

Thank you for your suggestions.

We added a paragraph in the introduction section. The design requirements of the robot for minimal invasive flexible surgery can be summarized into two conflicts. One is having enormous passive degrees of freedom (DoFs) to navigate through anatomical pathways and having sufficient controllable DoFs to operate on a remote surgical site. The other is being compliant to reduce damage to irrelevant tissues and being rigid to transmit force to operate the target. To solve these two conflicts, the proposed robot was installed on two nested hoses named the extension tube. The hoses were flexible to be bent while rigid to resist torsion. The proposed robot is actuated by the relative rotation of the two hoses, which occupy only one motor. Many bending modules attached to the tip of the robot were actuated by the time-share driven mechanism.

The design requirements concerning the medical application depend on specific scenarios. Several bending modules can be assembled in situ. The tip instruments can also be replaced. Specifically, for the proposed robot to exert in the stomach cavity to achieve transluminal diagnosis and treatment, the design requirements are as follows: the diameter of the largest part does not exceed 10 mm to pass through the oral cavity and esophagus (robot diameter 8 mm), the total length is longer than 80 cm and the extension tube is longer than 50 cm to pass through the incisor to the cardia, 4 DoFs of the manipulator (at least 2 bending modules) for complex operations, and no less than 1 Newton active load (bending force) to remove the foreign body.

-----Manuscript Revised (Introduction / Page 2)-----

Technically speaking, the stiffness of flexible devices is designed as a compromise between being stiff enough to enable distal propagation and operation and being compliant enough to adapt to the structural constraints along a tortuous path²³. To address this, it is required that the flexible robot reserves its shape and marches, stated as the follow-the-leader (FTL) motion²⁴. It was first proposed as an algorithm working on continuum robots with enormous DOFs. Lately, various FTL mechanisms²⁵ have been proposed in medical devices based on stiffening methods. To intrinsically follow the leader, two tubes take their turns to be the leader and get stiffened for the other to march forward²⁶. However, it remains unsettled to send the surgical instruments to the tip and enable its operations with sufficient DOFs. Our snake-inspired design fills up the last piece with a flexible rotation shaft. The flexible rotation shaft, metaphorized as a "vessel", can be extended to the very tip, and actuate endless BMs on the tip, thus providing as many DOFs as demanded.

Comment 6:

Lines 83-98 – it is very hard to understand what exactly the problem was here and how you solved it. I do not really understand it. What do you mean with “joint shift” and why exactly was it a problem?

Response:

Thank you for your comment. The bending motion of a continuum structure is approximated by discrete links dismembered by laser-profile technology (LPT) from a rigid stainless-steel tube. The joint shift is proposed to eliminate the bias in the approximation of the arc using discrete links. If the joints of discrete link have no clearance, the tip position of the discrete-link robot is not the same as that of the continuum robot. We fixed the arc length of the discrete-link structure with tendons. Then, the length of the discrete link is regarded as a variable to fit the arc of tendons. The length change of the discrete link should not exceed the clearance of laser-cutting, or the joint shift will not be realized.

The link length cannot be too short or the discrete links are too weak. The 4.5 mm is the maximum link length that the clearance of laser-cutting can tolerate. So, we adopted 4.5 mm link length for planar steerable pipe.

Fig. R2 Joint shift

For other tubes, we replaced the laser-profiled tube with a stainless-steel braided hose. The latter can also transmit rotation but contains no discrete link. We do not need to approximate the continuum structure anymore.

-----Supplementary materials (Fig. S1) -----

2. Details of the stainless-steel braided hose.

Fig. S1. Details of the stainless-steel braided hose. (A) The illustration of the stainless-steel braided hose. **(B)** The extension tubes. **(C)** The flexible inner pipe.

The stainless-steel braided hose contains three layers (Fig. S1A). The inner layer provides lubrication for inserting other shafts. The middle layer is a braided SS304 net. This layer resists torque and prevents buckling. The inner layer and outer layer are made of Polytetrafluoroethylene (PTFE), which is characterized by cold and heat resistance (180-260°C), acid and alkali resistance, resistance to various organic solvents, and its friction coefficient is very low, making it suitable as a coating for endoscopes that come into direct contact with human tissue.

Comment 7:

Lines 99-112: also rather hard to understand. The construction in Fig. 3 seems to be just a segmented steerable joint; it is unclear to me what the “state keeping” feature exactly is, or is it just an electric actuator that stops moving? Requires more explanation. Are the “super elastic” tendons made from SMA? Why not just steel cables?

Response:

Thank you for your comment.

- *it is unclear to me what the “state keeping” feature exactly is, or is it just an electric actuator that stops moving?*

The “state keeping” means that the screw pair does not rotate and the bending module holds its state. The idle bending module can hold a weight whether the motor is working to actuate other bending modules or not.

If the motor has to actuate all bending modules constantly to keep their states and resist external force, it would be impossible to actuate these bending modules one by one. Thus, “State keeping” is an important feature. Each bending module must hold its shape in the idle state when the motor is actuating other bending modules or shut down with no holding torque.

- *Are the “super elastic” tendons made from SMA? Why not just steel cables?*

The super elastic tendons are Ni-Ti alloy. It can recover from a large bending deformation. The elastic tendon can resist compression. We can both drag or push to steer the bending motions to two sides. If replaced with steel cables, we have to embed the cables on two sides of the planar steerable pipe, so that it can be steered to both sides. It would be hard to be actuated by the only screw pair.

Comment 8:

Lines 113-130: it is not clear to me how this principle works. What is the purpose of the “expansile tubular structure” in Fig. 4? What was the purpose of the SMA? What is exactly “selective” in this construction? The text zooms in into so much detail about some experiment with the SMA that the overall working principle is not well explained. “the SMA failed to drive a payload of 1000 g”. What was the required payload then? Why is 1 Kg not enough?

Response:

Thank you for your comment. This section basically introduces a clutch actuated by SMA.

- *What is the purpose of the “expansile tubular structure” in Fig. 4?*

The expansile tubular structure aims to control the connection between the screw pair and the flexible inner pipe. It is an elastic structure driven by the SMA wire. The structure gets larger in the radial direction if compressed in the axial direction. The elastic structure is placed inside the hollow screw pair. The expansion in the radial direction

makes the structure attached to the screw pair.

- *What was the purpose of the SMA?*

The SMA is to control the compression of the expansile tubular structure in the axial direction. When the SMA is heated by electricity, it contracts and applies a compression force on the structure.

- *What is exactly “selective” in this construction?*

We can select the bending module by activating its SMA clutch. If the SMA clutch is activated, this bending module is able to be actuated by the motor. The rest of the bending modules are in an idle state and hold their shapes.

- *What was the required payload then? Why is 1 Kg not enough?*

The payload is important because a friction force between the screw pair and the expansile tubular structure actuates the screw pair. The friction force is related to the compression force that the SMA wire can generate. As for the required payload, it's quite hard to evaluate. The setting of 1 Kg was based on our experimental test.

We have found that the slipping happens if the friction force is insufficient, while it is hard to measure the contact state from outside. We decided to revise the design of the SMA clutch. The performance of the new SMA clutch is demonstrated in Movie. S4.

-----Manuscript Revised (Results / Page 4~5)-----

To test the SMA clutch, a 4.0 V voltage input was plugged in that lasted 0.2 s, followed by the PWM input with 6% duty. The parameters were suggested by the results of the SMA wire test. As the result shown in Fig. 3F and Movie S4, the toothed slider locked the toothed screw in 0.1 s. During the power-on period (10 s), the toothed slider was firmly attached to the toothed screw. It took another 2 s to unloose the connection (Movie S4).

Fig. 3. Shape memory alloy works as a clutch for selective driving each BM. (F) Snapshots of the SMA clutch in its working progress. The SMA clutch was actuated in the 2.0 s. The power was cut off in the 12 s. The toothed slider unloosed the Toothed screw in the 14s.

Comment 9:

Fig. 4B shows several plots but it is unclear to me what these plots mean and what I can conclude from the figure. Why do you show these plots? What was the design requirement exactly and how can I see from Figs 4A and 4B what solution is best?

Response:

Thank you for your comment. The aim is to shorten the response time while maintaining the contraction state until the PWM pulse is finished. **The best solution is to start with a full-duty PWM input and switch to a low-duty PWM input to reduce energy consumption.**

We already recognized that the temperature related to the displacement of the SMA wire should also be taken into consideration. We designed several tests to analyze the relationship between the temperature and the displacement on a homemade testing table. The best solution concerning **voltage, pulse duration, and duty of the PWM**. These factors were analyzed in the revised manuscript.

-----Manuscript Revised (Results / Page 4~5)-----

Performance of the SMA clutch

The material for selective stimulation is a kind of SMA^{32,33} – nitinol. The contraction of nitinol wire occurs as the temperature changes³⁴. The SMA wire with a diameter of 150 μm suggested by Kim et al³⁵ was adopted because of its shorter cooling process and splendid durability. To design a strategy to control the SMA wire and validate its performance, a homemade sliding table (the left picture in Fig. 3A) was designed. The SMA wire contracted and dragged a sliding block against a tension spring (Movies S3). The displacement of the sliding block was measured by a micrometer, and the temperature was recorded by a thermal camera (the right figure in Fig. 3A). The SMA wire was heated by a pulse with different voltage inputs and pulse duration. The maximum displacement and corresponding temperature were recorded in Fig. 3B. The displacement started when the SMA wire was heated to 40 °C. No further contraction when the temperature of the SMA wire exceeded 50 °C. The results were summarized from three aspects. First, the maximum displacement was around 1.6 mm. The percentage of the contraction was solved to be 4% (The effective length of the SMA wire was 40 mm.). Second, a larger voltage input shortened the needed pulse duration to reach maximum displacement due to larger heating power. Third, the pulse should be halted when the temperature reaches 50 °C because it only increases the energy consumption. Also, high temperature was a danger in medical applications.

Fig. 3. Shape memory alloy works as a clutch for selective driving each BM. (A) Left: the setting for the SMA wire test; Right: the thermal map when the SMA wire was working. (B) The maximum displacement and temperature of the SMA wire with different durations of pulse input. The data contain three groups with voltage changing from 3.0 V to 5.0 V. (C) The temperature changes when heated by PWM inputs with different duties. The duties of the PWM inputs were 2%, 4%, 6%, 8%, and 10%. In each trial, the SMA wire was activated by a 4.0 V input at 2.0 s which lasted 0.2 s, then powered by the PWM inputs with different duties. The power was cut off in the 12 s and the cooling process was also recorded. (D) The displacement of the SMA wires in the test with different duties. (E) The illustration of the SMA clutch. When heated, the SMA wire contracted and pulled the toothed slider to the left. Thus, the tooth slider was locked with the toothed screw. Then, the toothed screw started to rotate with the flexible inner pipe. When the SMA wire cooled down, an embedded compression spring pressed the toothed slider to its origin state. (F) Snapshots of the SMA clutch in its working progress. The SMA clutch was actuated in the 2.0 s. The power was cut off in the 12 s. The toothed slider unlocked the Toothed screw in the 14s. (G) The thermal map when the SMA clutch was working at room temperature (28 °C). (H) The thermal map when the SMA clutch was working at 39 °C.

Pulse width modulation (PWM)³⁶ was introduced to reduce energy consumption. The voltage input was switched to a PWM input with five duties to maintain the contracted state. Among the five trials, the one with 4% duty maintained a relatively low temperature (40 °C) (Fig. 3B). The displacements of the five trials were recorded in Fig. 3D. The SMA wire reached the maximum displacement in a few milliseconds. Those trials that kept the temperature over 40 °C maintained the maximum contraction until the PWM input was stopped (Only the trial with 2% duty failed). The SMA with 4% duty experienced a shorter recovery time back to the original length. These tests suggested that we could adjust the voltage, pulse duration, and duty of the PWM to promote the contraction performance of the SMA wire.

To control the connection between the flexible inner pipe and the screw pair, the SMA wire was folded and hooked to a toothed slider as illustrated in Fig. 3E. The two ends of the SMA wire were pinned on a fixed ring and connected to two wires for power supply. The screw pair contained a toothed screw for the toothed slider to bite in. The tooth slider got released by a compression spring hidden between the toothed screw and the toothed slider. This movable structure was named the SMA clutch. Six more wires were embedded in the flexible inner

tube to activate subsequent SMA clutches. Two ends of each wire were connected to the pins of a female plug and a male plug respectively (Fig. S3 in the Supplementary materials).

To test the SMA clutch, a 4.0 V voltage input was plugged in that lasted 0.2 s, followed by the PWM input with 6% duty. The parameters were suggested by the results of the SMA wire test. As the result shown in Fig. 3F and Movie S4, the toothed slider locked the toothed screw in 0.1 s. During the power-on period (10 s), the toothed slider was firmly attached to the toothed screw. It took another 2 s to unloose the connection (Movie S4).

Comment 10:

Was the experiment in the stomach carried out by a clinician in a clinical setting?

Response:

The diagnosis and therapy experiments were performed by one clinician with extensive experience in gastroscopy in the Microanatomy Laboratory of the Second Affiliated Hospital of Zhejiang University School of Medicine.

Comment 11:

The discussion is very technical, add a section about the clinical applicability of the design and issues still to be solved for clinical application, for example dealing with sterilisation, patient safety, biocompatibility, etc.

Response:

Thank you for your comment. Patient safety is the most critical aspect of medical instrument design. In consideration of the temperature change of the manipulator during operation, we add a section describing the performance of the SMA clutch. The highest temperature was 30.8 °C detected on the wire connected to the SMA wire directly in ambient air at 26 °C. In a simulated human body temperature environment, the SMA clutch works well and does not cause additional temperature increases.

For the therapy experiment, the inner and outer layers of the extension tube were composed of polytetrafluoroethylene, which is acid-resistant, self-lubricating, and harmless to biological tissues. Polyurethane membranes were used to encapsulate the BMs, which are biocompatible and flexible, making the manipulator waterproof. Normative and proper encapsulation is necessary for the proposed robot to be applied in clinical practice. The currently used encapsulation of the gastroscopy system is also suitable for the robot, which integrates the robot and other functional channels and gives the robot biocompatibility to ensure safe contact with human tissue and to meet the requirements of internal channel sterilization and external surface sterilization.

We added a section on the design of clinical applicability to the discussion and a section on the issues that needed to be solved for clinical application, such as the integration of more channels for suction, inflation, and flushing.

-----Manuscript Revised (Discussion / Page 8~9) -----

Further, we tested the prototype's performance in clinical application scenarios, including a diagnosis experiment on a human stomach model and a therapy experiment on a porcine stomach. The robot is capable of passing through narrow openings and along tortuous paths toward remote operating sites in Movies S6 and S7, satisfying the operation for transluminal diagnosis. In addition, the robot reveals excellent flexibility and maneuverability in the transluminal therapy experiment, as compared to the gastroscopes, the robots could

approach the fishbone at an optimal angle and lift it, rather than dragging the bone along the endoscope axis as in gastroscopic procedures, minimizing the surgery-related damage to the tissue. These results showed the potential of our robot to be a new solution to handle multi-DOF tasks in difficult-to-reach surgical spots.

In the future, it is worth embedding sensors in BMs for motion feedback so that closed-loop control can be established for robot-assisted surgery. Meanwhile, motion efficiency can be promoted. The idea of a time-share driven mechanism sacrificed time efficiency in switching between BMs. It was proved to be feasible that the BMs could move at individual paces to the planned bending angle while the motor kept running to control the speed of movement. Last but not least, although manual control was involved in the diagnosis and therapy experiments, robotic control and tip navigation are feasible under the established kinematic model of a time-share driven robot (Fig. S8 in the Supplementary materials). For clinical applications of the proposed robot, it needs to integrate more channels for suction, inflation, and flushing. Those channels can surround the robot and form a robotic endoscope system with a tube diameter of 1cm using the encapsulation of currently used gastroscopy. This gives the robot biocompatibility to ensure safe contact with human tissue and to meet the requirements of internal channel sterilization and external surface sterilization. The time-share driven mechanism provides an unprecedented way to bring the controllable DOFs and the flexibility of the continuum robot to the next level.

REVIEWERS' COMMENTS

Reviewer #1 (Remarks to the Author):

The revised manuscript has addressed my concerns and is acceptable now.

Reviewer #2 (Remarks to the Author):

Many of the comments have been diligently addressed in the revised manuscript. The authors have developed a new prototype and conduct new experiments to incorporate the suggested changes. I am confident that the current version is ready for publication in Nature Communications.

Reviewer #3 (Remarks to the Author):

General: the English is quite OK, but needs still here and there some polishing

Do not use abbreviations in the figure captions; figure captions should be readable separately from the text.

The construction is more clear now but at some points still quite confusing, especially in the added yellow texts, see remarks below. The paper still gets somewhat lost in detail while not being clear about the overall functioning of the design. It would be good to add a very clear exploded view of ALL the components of the design, add in this exploded view all the names and numbers, and CONSISTENTLY use these exact same names without exception everywhere in the text (same thing same name).

It would be better to replace the word "pipe" by "tube" because a pipe suggests that it is rigid whereas a tube is flexible, which is the case here. The word "pipe" is confusing.

Lines 89 – 90: is the "hose" and the "tube" the same thing as the "pipe" mentioned earlier? Use then the same word to avoid confusion.

Line 106: Why is the "steerable pipe" planar? The system works in 3D no, not in a 2D plane. Or does the snake-like system only work in a 2D plane?

Lines 106-107: So is this "thin-walled stainless steel tube containing a series of interlocked segments" the same component as the earlier mentioned "steerable pipe" or "steerable tube"? Or is it another component?

Lines 110-111: The BM is assembled with one "flexible inner pipe" sealed inside one "planar steerable pipe". Again the confusion whether this is the earlier mentioned hose, tube, or another component.

Lines 111-112: "the bending behaviour is similar to other continuum designs". Similar in what aspects?

Lines 118-154: These lines contain lots of details on the used SMA as a component in the clutch, but it is not mentioned how exactly the clutch works. Explain also the other components of the clutch and how these parts collaborate together with the SMA wire.

Response Letter

Reviewer #1

General Comment:

The revised manuscript has addressed my concerns and is acceptable now.

Response:

Thank you for your valuable advices.

Reviewer #2

General Comment:

Many of the comments have been diligently addressed in the revised manuscript. The authors have developed a new prototype and conduct new experiments to incorporate the suggested changes. I am confident that the current version is ready for publication in Nature Communications.

Response:

Thank you for your valuable advices.

Reviewer #3

General Comment:

the English is quite OK, but needs still here and there some polishing.

Comment 1:

Do not use abbreviations in the figure captions; figure captions should be readable separately from the text.

Response:

Thank you for your suggestion. We've refrained from utilizing abbreviations in figure captions. The abbreviations presented in figures are defined in the associated legends.

Comment 2:

The construction is more clear now but at some points still quite confusing, especially in the added yellow texts, see remarks below. The paper still gets somewhat lost in detail while not being clear about the overall functioning of the design. It would be good to add a very clear exploded view of ALL the components of the design, add in this exploded view all the names and numbers, and CONSISTENTLY use these exact same names without exception everywhere in the text (same thing same name).

Response:

The core functionality of the entire design hinges upon the bending module. In Fig. 1B, we have modified the depiction to reveal the interrelation between the planar steerable tube, flexible inner tube, and the SMA clutch. This revised figure elucidates how the rotation input translates into the bending output of this one-degree-of-freedom (DoF) bending module.

For a comprehensive view of all design components, an exploded representation is provided in the supplementary materials (refer to Fig. S2-S4). Due to spatial limitations, Figure 3 and Movie S4 offer detailed insights specifically centered on the SMA clutch. This element stands as a pivotal component, facilitating the time-share driven mechanism. We meticulously cross-verified the consistent usage of names throughout several review rounds.

Fig. 1. Schematic illustration of the proposed bioinspired time-share driven robot. (A) Design of the snake-inspired time-share driven robot. **(B)** Schematic of one bending module. The screw pair, locked to the shape memory alloy (SMA) clutch, facilitates the conversion of rotational motion into translation, manipulating the elastic tendon to steer this bending module. **(C)** Illustration of the active steerability of a handheld time-share driven continuum robot within a human stomach.

Comment 3:

It would be better to replace the word “pipe” by “tube” because a pipe suggests that it is rigid whereas a tube is flexible, which is the case here. The word “pipe” is confusing.

Response:

Apologies for the confusion. The term 'pipe' has been replaced with 'tube'.

Comment 4:

Lines 89 – 90: is the “hose” and the “tube” the same thing as the “pipe” mentioned earlier? Use then the

same word to avoid confusion.

Response:

Clarification on our terminology is crucial. Our design incorporates several similar concepts. We distinguish between the planar steerable tube (tube 1) and the flexible inner tube (tube 2) as elements of the bending module. The term 'tubes' is employed, emphasizing their flexibility.

The 'passive' extension tube (tube 3) is incapable of active bending and comprises the inner hose (hose 1) and outer hose (hose 2), which are distinguished by the high slenderness ratio (the length divided by radius).

Additionally, the flexible inner tube (tube 2) contains a section of passive and flexible hose (hose 3) to transmit rotation in the curved shape of the planar steerable tube.

At last, the inner hose (hose 1) of the passive extension tube (tube 3) connects all the flexible inner tubes (tube 2), aiming to transmit the rotation along the robot. We termed flexible rotation shaft (shaft 1) to emphasize its function.

Comment 5:

Line 106: Why is the “steerable pipe” planar? The system works in 3D no, not in a 2D plane. Or does the snake-like system only work in a 2D plane?

Response:

The term 'planar' in 'planar steerable tube' denotes its single degree of freedom, enabling bending solely in a 2D plane. However, when connected in varying configurations, the snake-like system can operate in a 3D plane, harnessing the multiple degrees of freedom conferred by the assembled bending modules.

Comment 6:

Lines 106-107: So is this “thin-walled stainless steel tube containing a series of interlocked segments” the same component as the earlier mentioned “steerable pipe” or “steerable tube”? Or is it another component?

Response:

The interlocked segments are part of planar steerable tube. I would call the interlocked segments group “tube” because they are cut from the same thin-walled stainless tube and they keep matched with adjacent segments. This tube incorporates a rigid section housing the SMA clutch (as depicted in the revised Fig. 1B). Determining a precise nomenclature is challenging; however, the bending module essentially comprises two layers: the planar steerable tube and the flexible inner tube. The SMA clutch operates between the rigid sections of both layers (see Fig. 1B).

Comment 7:

Lines 110-111: The BM is assembled with one “flexible inner pipe” sealed inside one “planar steerable pipe”. Again the confusion whether this is the earlier mentioned hose, tube, or another component.

Response:

Please refer to the response in question 4, it concerns tube 1 and tube 2 within the bending module.

Comment 8:

Lines 111-112: “the bending behaviour is similar to other continuum designs”. Similar in what aspects?

Response:

We revised the sentence.

.....

The manipulation of the planar steerable tube within a single BM was accomplished through tendons, a method analogous to the operational principle found in other tendon-driven continuum robots²⁸⁻³¹.

Comment 9:

Lines 118-154: These lines contain lots of details on the used SMA as a component in the clutch, but it is not mentioned how exactly the clutch works. Explain also the other components of the clutch and how these parts collaborate together with the SMA wire.

Response:

We have provided further insight into the SMA clutch's constituent parts, such as the toothed slider and screw pair, elucidating its operational mechanism. The revised paragraph below offers clarity on the SMA clutch's functionality.

-----Manuscript Revised (Results / Page 5)-----

The functionality of the SMA clutch relies on the SMA wire to toggle the connection between the flexible inner tube and the screw pair, as depicted in Fig. 3E. The SMA wire was folded and securely hooked to a toothed slider. Its two ends were anchored to a fixed ring and linked to two power supply wires. Within the screw pair, a toothed screw facilitated engagement with the toothed slider. The toothed slider was released by a compression spring concealed between itself and the toothed screw, forming a mobile structure referred to as the "SMA clutch". In addition, six more wires were embedded in the flexible inner tube to activate subsequent SMA clutches. Each wire's two ends were connected respectively to the pins of a female plug and a male plug (refer to Fig. S3 in the Supplementary materials).